# LayerCake: Token-Aware Contrastive Decoding within Large Language Model Layers

## Abstract

Large language models (LLMs) excel at natural language understanding and generation but remain vulnerable to factual errors, limiting their reliability in knowledge-intensive tasks. While decoding-time strategies provide a promising efficient solution without training, existing methods typically treat token-level and layer-level signals in isolation, overlooking the joint dynamics between them. In this work, we introduce a token-aware, layer-localized contrastive decoding method that aligns specific token types with their most influential transformer layers to improve factual generation. Through empirical attention analysis, we identify two key patterns: punctuation tokens receive dominant attention in early layers, while conceptual tokens govern semantic reasoning in intermediate layers. By selectively suppressing attention to these token types at their respective depths, we achieve the induction of controlled factual degradation and derive contrastive signals to guide the final factual decoding. Our method requires no additional training or model modification, and experiments demonstrate that our method consistently improves factuality across multiple LLMs and various benchmarks.

## 1 Introduction

Large language models (LLMs) have recently achieved remarkable performance across a broad range of natural language understanding and generation tasks, from open-ended dialogue to factual question answering OpenAI Group (2023); Touvron et al. (2023a); Anil et al. (2023); Bai et al. (2023). However, despite their fluency and coherence, LLMs often generate content that deviates from factual reality. To mitigate this issue, commonly referred to as *hallucination* Ji et al. (2023); Huang et al. (2023a); Zhang et al. (2023), previous efforts have primarily focused on data filtering and augmentation Abbas et al. (2023); Gunasekar et al. (2023); Touvron et al. (2023b), model editing Dai et al. (2022); Huang et al. (2023b); Mitchell et al. (2022), and model architecture design Li et al. (2023a); Liu et al. (2023a;b). Recently, decoding-time strategies have emerged as a promising a training-free, architecture-agnostic alternative Chuang et al. (2024); Zhang et al. (2024); Wu et al. (2025). To make decoding operations more effective in anti-hallucination, a key open question is how to leverage the internal signals of LLMs to guide generation toward factual accuracy.

Existing works leverage internal model signals from two perspectives. At the layer level, intermediate layers are recognized as the stage where the model begins to interpret factual semantics and perform reasoning, indicated by distinct early-exit logits or hidden-state representations Wang et al. (2025); Chuang et al. (2024); Skean et al. (2025). At the token level, specific tokens, such as beginning-of-sequence markers, receive significantly higher attention than others, and such attention patterns are essential for maintaining factual output generation Barbero et al. (2025). These findings suggest that even when LLMs generate hallucinated outputs, they internally encode truthful signals that can be retrieved through well-targeted decoding strategies. However, existing decoding-time methods are designed to utilize either layer-wise dynamics or token-specific signals in isolation. This separation limits their ability to fully recover reliable factual evidence during generation. To address this limitation, we propose a decoding strategy that jointly considers the functional specialization across LLM layers and the semantic roles of key token types, aiming to better align generated content with the factual knowledge embedded in the model.

Considering the distinct functional roles that different layers and token types play in factual reasoning, we analyze how attention is distributed across layers with respect to specific token categories. Our

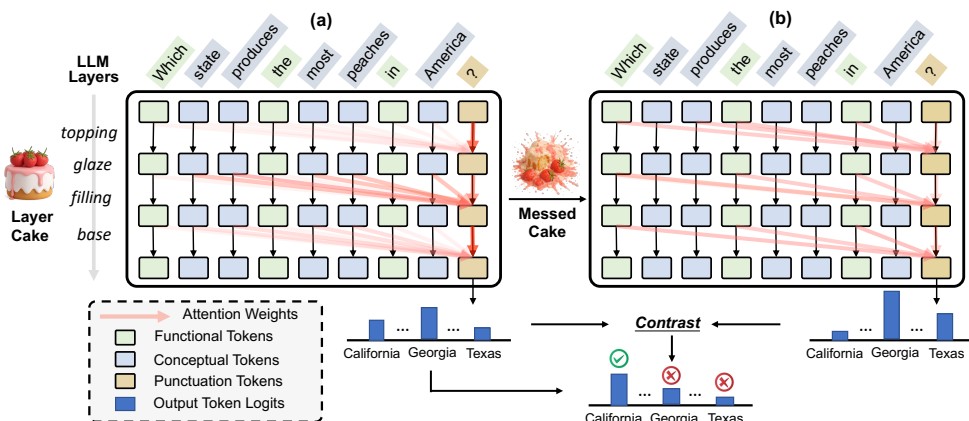

Figure 1: We conceptualize LLMs as a layered cake, where different layers contribute distinct "flavors" to the reasoning process. (a) We show the attention of the first predicted answer token over input question tokens across layers. Early layers (the "topping") focus on structural elements, attending strongly to punctuation and special symbols. Intermediate layers (the "glaze") shift attention to conceptual tokens that carry semantic meaning and support factual reasoning. Functional tokens consistently receive low attention, indicating limited contribution to content understanding. (b) We suppress attention to specific token types at their most influential layers, creating a perturbed reasoning trajectory. By comparing original and perturbed outputs, we derive contrastive logits that guide decoding toward more factually aligned predictions.

analysis reveals two dominant patterns: 1) punctuation tokens, such as beginning-of-sequence markers and commas, receive substantial attention in the early layers, shaping the model's initial structural alignment; and 2) conceptual tokens, which carry the core semantic content of the input, attract increasing attention in intermediate layers, where factual reasoning typically occurs. In contrast, remaining tokens such as "what" and "the" receive relatively low attention across all layers, reflecting their functional rather than semantic role in the reasoning process. This progression resembles a layered cake, where each layer contributes a distinct "flavor" to the generation process (Figure 1). In this view, different token types play their most critical roles at specific depths rather than being uniformly important throughout the network.

In contrast to prior decoding-time methods that treat layer-level and token-level signals in isolation, our approach explicitly links each token category to its most influential layer range and applies targeted attention suppression. As illustrated in Figure 1, we intervene on the attention assigned to functional or conceptual tokens at their respective layers to simulate a "messed" reasoning process. We then contrast the perturbed predictions with the original ones to expose the factual contributions of each layer-token interaction. To recover truthful outputs, we compute contrastive logits by subtracting the perturbed predictions from the original ones. This contrast identifies generation steps that are particularly sensitive to factual disruptions, enabling decoding to be guided by layer-aware token contributions that are most relevant for factual alignment.

We conduct extensive experiments on LLaMA 2 Touvron et al. (2023b), LLaMA 3 Dubey et al. (2024), Mistral Jiang et al. (2023) and Qwen 3 Yang et al. (2025) spanning several benchmarks including TruthfulQA Lin et al. (2021), FACTOR (Expert)Muhlgay et al. (2023), HellaSwagZellers et al. (2019), StrategyQA Geva et al. (2021), GSM8K Cobbe et al. (2021), HaluEval-Sum Li et al. (2023b) and NQ Kwiatkowski et al. (2019) Results demonstrate that our approach achieves state-of-the-art performance in reducing hallucinations, without the need for additional training or architectural modifications.

The main contributions of this work are:

• We introduce **LayerCake**, a decoding-time framework that bridges transformer layer depth with token semantics by aligning different token types to the specific layers where they play the most critical roles in factual reasoning.

- We conduct a fine-grained attention intervention analysis that reveals how factual accuracy depends on both the type of attended token and the depth at which it is processed, uncovering token-layer interactions that are essential for faithful generation.
- We design a contrastive decoding strategy that leverages these layer-token insights to guide generation through targeted attention modulation. Our method achieves strong factuality gains across multiple LLMs and benchmarks, without any additional training or architectural modifications.

## 2 RELATED WORK

**Decoding Improvement for LLMs.** Recent work has introduced a variety of inference-time decoding strategies aimed at improving the factuality and reliability of large language models (LLMs) without requiring additional training. A shared assumption underlying many of these methods is that different layers within LLMs serve distinct functional roles during generation: early layers primarily encode surface-level patterns and structural input features (Azaria and Mitchell, 2023; Wang et al., 2024; Dong et al., 2022), middle layers capture semantically rich abstractions essential for reasoning and factual grounding (Wang et al., 2025; Jin et al., 2025; Liu et al., 2025; Skean et al., 2025), and final layers are mainly responsible for fluent token-level prediction at the vocabulary level (Chuang et al., 2024; Zhang et al., 2024; Liu et al., 2024). Building on this layer-wise functional view, contrastive decoding methods aim to extract more faithful or informative signals by comparing model predictions across alternative decoding paths. These comparisons span expert and weaker models to emphasize reliable outputs (Li et al., 2023c; Zhang et al., 2025), internal representations between early and final layers of the same model (Chuang et al., 2024; Zhang et al., 2024), or contrastive signals induced through hallucination-triggering perturbations, such as attention dispersion or input corruption (Jiang et al., 2025; Kai et al., 2024; Zhang et al., 2025).

Beyond contrastive strategies, other approaches operate directly on internal model representations during inference. Some manipulate hidden states along activation directions associated with truthful outputs (Li et al., 2023d; Zou et al., 2023; Jorgensen et al., 2024; Stoehr et al., 2024), while others identify key evidence spans via attention analysis to revise the input prompt (Liu et al., 2025), or use flow-based transformation vectors to align hallucinated states with truthful representations (Wang et al., 2025). In parallel, confidence-guided methods adjust output distributions based on signals such as contextual entropy (Chen et al., 2024a), sharp cross-layer probability growth (Wu et al., 2025), or test-time activation clipping to suppress overconfident hallucinations (Chen et al., 2024b). While prior work focuses either on inter-layer dynamics or token-level manipulations, our approach bridges the two. We study how specific token types contribute to factual reasoning at different depths of the model, and use this token-layer correspondence to induce contrastive signals.

**Not All Tokens Are Equal in LLMs.** Recent studies have consistently suggested that not all tokens contribute equally to the behavior of large language models (LLMs), highlighting the need to treat tokens differently during both training and inference. Some methods show that selectively supervising task-relevant tokens during training improves alignment and generalization, either by explicitly identifying important tokens or by applying dropout to uninformative ones (Lin et al., 2024; Hans et al., 2024). Others perform token-level weighting or masking to focus learning signals on alignment-relevant positions, reducing redundancy and enhancing efficiency (Thankaraj et al., 2025; Christopoulou et al., 2024). At inference time, recent works reveal that language models often over-allocating focus to certain tokens, such as the first position, which serves as an attention sink and strongly influences model predictions (Barbero et al., 2025; Gu et al., 2025). These findings establish a foundation for our work: token contributions are unequal and context-dependent. Instead of treating token importance as static or uniform, we analyze how different token categories exert varying influence across model depth. This token-layer interaction offers a structured view of internal model dynamics, which we leverage to guide factual generation through layer-aware decoding interventions.

## 3 TOKEN-LEVEL ATTENTION BEHAVIOR ACROSS LLM LAYERS

Understanding how Large Language Models (LLMs) allocate attention across different input tokens provides crucial insights into their internal reasoning process. We analyze attention patterns to identify token types that play distinctive roles across different layers, laying the foundation for our decoding intervention strategy. Our analysis is based on both case-specific and statistical evidence.

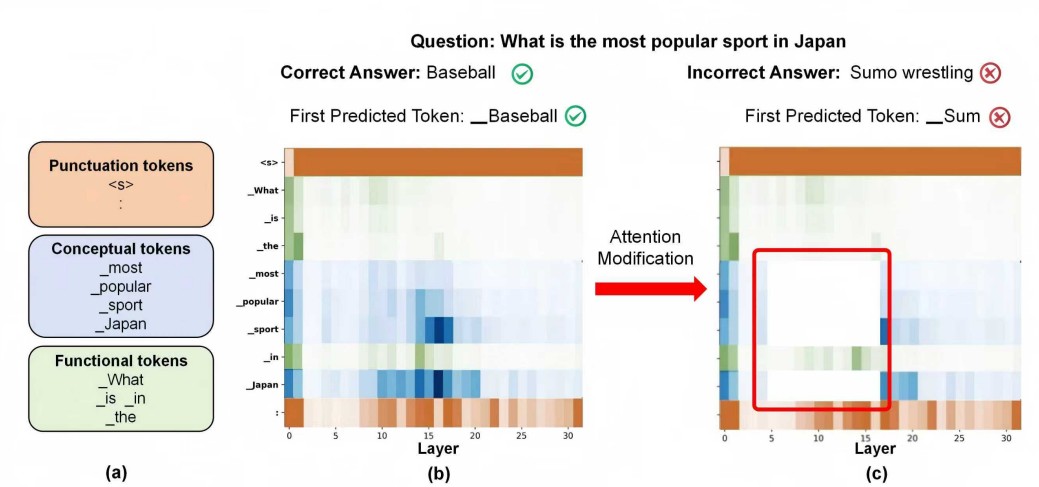

Figure 2: We analyze the attention behavior of the first predicted token using a TruthfulQA question answer pair: <"What is the most popular sport in Japan:", "Baseball">. **(a)** Input tokens are categorized into three groups and each category is visually highlighted with different colors. **(b)** The heatmap illustrates the attention distribution from the first predicted token ("Baseball") to all input tokens across 32 layers, with darker shades indicating stronger attention. **(c)** We conduct targeted attention suppression by masking punctuation/conceptual tokens in early/middle (highlighted in red), respectively. This intervention disrupts the model's original reasoning process, leading to an incorrect prediction ("Sumo") instead of the correct answer ("Baseball"), underscoring the importance of token-layer interactions in factual prediction.

## 3.1 CASE STUDY

We begin with a representative example from the TruthfulQA Lin et al. (2022): "What is the most popular sport in Japan:" This question has a clear answer ("Baseball"), making it suitable for analyzing how attention supports factual reasoning. In Figure 2, we examine how the model allocates attention when predicting the first output token, reflecting its initial focus during factual generation.

## 3.2 STATISTICAL ANALYSIS

To validate and generalize the attention behaviors observed in our case study, we perform a statistical analysis over the full TruthfulQA dataset Lin et al. (2022). For each question, we categorize input tokens into three types: punctuation tokens (P), conceptual tokens (C), and functional tokens (F). The classification is based on part-of-speech tags and semantic roles, following the definitions in Section 3.1.

We use 32-layer LLaMA 2-7B model Touvron et al. (2023b) as an example and extract attention weights from the first output token of the generated answer. For each layer, we measure the attention assigned to each input token and compute the total proportion of attention allocated to each token type. We then aggregate these proportions across all questions to obtain an average layer-wise distribution per token category. Figure 3 (a) presents the results of this analysis. The $x$-axis denotes the layer index, and the $y$-axis represents the average proportion of attention received by each token type. We observe several trends and patterns emerge from the distribution:

- In the early layers (0 to 4), punctuation tokens receive the highest attention. This suggests that the model initially focuses on structural elements, including the beginning-of-sequence marker  and punctuation symbols.
- From Layer 5 to 16, the attention gradually shifts toward conceptual tokens, reflecting the model's semantic grounding and reasoning process.
- Between Layers 17 and 27, the attention assigned to conceptual tokens declines, indicating a stage of information consolidation rather than new semantic extraction.
- In the final layers (28 to 31), attention to conceptual tokens increases again. This reflect a final alignment phase, where the model revisits key semantic elements before generating the output.

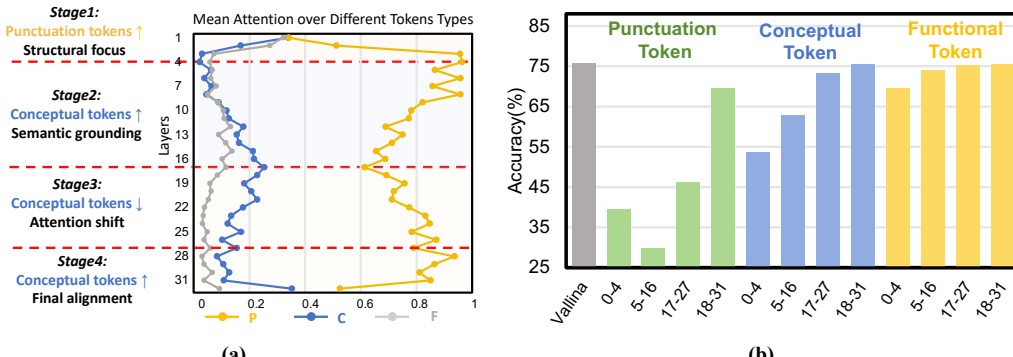

**(a)**                                **(b)**

Figure 3: (a) We plot the average proportion of attention received by punctuation, conceptual, and functional tokens from the first predicted answer token, aggregated across TruthfulQA. Punctuation (P) tokens dominate in early layers, conceptual (C) tokens receive increasing attention in the middle layers, and functional (F) tokens remain consistently low. (b) Effect of attention suppression across layers and token types. We report accuracy drops on HellaSwag when suppressing attention to punctuation (P), conceptual (C), or functional (F) tokens at different layer stages of LLaMA 2.

These results confirm that attention is not uniformly distributed across token types or layers. Instead, the model processes different types of tokens at distinct stages, with punctuation tokens guiding early structural encoding, conceptual tokens supporting semantic reasoning in the middle layers, and both types influencing the final prediction. This structured attention progression supports our motivation to design layer-aware decoding strategies that intervene based on token type and layer position.

## 3.3 TOKEN-SPECIFIC INTERVENTION ACROSS LLM LAYERS

Building on the attention patterns identified in Section 3.1 and 3.2, we design a token-type-aware intervention experiment to quantify how specific tokens contribute to factual generation at different layers of LLMs. Rather than treating attention as a monolithic mechanism, we aim to disrupt it in a targeted manner by selectively suppressing attention to specific token types (Punctuation, Conceptual, and Functional) at the stages where they are most influential. This setup allows us to isolate the causal role of each token category across the model's reasoning trajectory.

We divide the 32-layer LLaMA 2-7B model into four stages based on the attention flow: structural encoding (Layers 0–4), where punctuation tokens dominate; semantic grounding (Layers 5–16), where conceptual tokens receive increasing focus; semantic consolidation (Layers 17–27), where attention becomes more diffuse; and final prediction (Layers 28–31), where key semantic elements are revisited for output generation. These stages serve as natural boundaries for controlled layer-wise intervention.

To perturb the attention mechanism, we adopt a suppression-based strategy that eliminates the contribution of a selected token type at specific layers. This approach avoids the instability often introduced by attention amplification and enables precise manipulation without modifying model parameters. Specifically, for each decoding step, we retrieve the attention map $\tilde{A}^l_{n,j}$ from layer $l$, where $n$ is the position of the output token and $j$ indexes input tokens. We then assign $-\infty$ to the attention logits of the targeted token type before softmax normalization:

$$\tilde{A}^l_{n,j} = \begin{cases} -\infty, & j \in S_T \text{ for } T \in \{P, C, F\}, \\ \tilde{A}^l_{n,j}, & \text{otherwise.} \end{cases} \quad (1)$$

$S_T$ denotes the set of indices for token type $T \in$ (Punctuation, Conceptual, Functional). After modification, the logits are re-normalized to yield a valid attention distribution. We apply this perturbation across the four stages and three token types on HellaSwag, and report the performance degradation in Figure 3 (b). We have three key findings: 1) Early-stage intervention has the most impact, emphasizing the importance of structural and semantic grounding; 2) Punctuation suppression leads to larger performance drops, possibly due to the disruption of attention anchoring mechanisms such as the  token; 3) Functional tokens have minimal impact when suppressed, consistent with

their limited semantic role. The results confirm that token-type and layer-level contributions to factual reasoning are both non-uniform and tightly coupled.

# 4 METHOD: TOKEN-AWARE CONTRASTIVE ATTENTION DECODING

Inspired by the observations in Section 3, we find that attention interventions on different token types at various layers can have differing impacts on the model's final predictions, but they consistently lead to a degradation in performance. This suggests that such interventions can disrupt the model's internal processes at different stages and thus serve as a mechanism to induce hallucinations. Building on this, we adopt a contrastive decoding approach to suppress potential hallucinated outputs. Specifically, this involves comparing the token distributions of the base model and the intervened model, and reweighting the next-token probabilities of the base model accordingly:

$$p\left(x_t \mid x_{<t}\right) \propto \exp\left[(1 + \alpha) \log p_{\text{ori}}\left(x_t \mid x_{<t}\right) - \log p_{\text{mod}}\left(x_t \mid x_{<t}\right)\right]. \tag{2}$$

In Equation 2, a new next-token distribution $p(x_t|x < t)$ is derived by comparing the next-token distribution of the original model $p(x_{ori}|x < t)$ and that of the intervened model $p(x_{mod}|x < t)$. A scaling factor $\alpha \epsilon R$ controls the relative influence between the original and the intervened models. When $\alpha > 0$, the distribution emphasizes the probabilities from the original model, leading to a preference for token predictions that align with its output.

Although we found that attention interventions in the earlier layers are more sensitive and more likely to cause performance degradation and undesirable outcomes, applying contrastive decoding remains a challenging task. We aim to determine the most appropriate contrastive strategy at each stage by understanding how the model allocates attention to the input throughout its processing phases.

## 4.1 PUNCTUATION TOKENS IN THE EARLY STAGE

For the structural encoding stage interval, based on the analysis in Section 3, the model undergoes a transition from globally gathering information to a stage where the sink effect intensifies and little attention is given to additional tokens. Thus, reducing attention to P-type tokens directly prompts the model to attend more to other tokens. We prefer an over-attention bias toward concept-related tokens, as these C-type tokens are more likely to trigger hallucination issues, which benefits the effectiveness of subsequent contrastive decoding. Based on the above analysis, we introduce a threshold parameter $th_a$ to suppress attention being excessively assigned to P-type tokens in the first stage $l \in L_{\text{stage1}}$:

$$\tilde{A}_{n,j}^l = \begin{cases} -\infty, & \tilde{A}_{n,j}^l > th_a \text{ and } j \in S_P \\ \tilde{A}_{n,j}^l, & \text{otherwise} \end{cases} \tag{3}$$

## 4.2 CONCEPTUAL TOKENS IN THE MIDDLE STAGE

For the semantic grounding stage interval, the model progressively attends more to semantically relevant tokens and the straightforward way to induce hallucinations is to suppress the attention to conceptual tokens. We introduce a threshold parameter $th_b$ for suppressing attention to semantically related tokens, which is determined by the total attention weight assigned to all conceptual tokens at a given layer, enabling us to adaptively identify layers $l \in L_{stage2}$ where the model begins focusing on semantic information.

$$\tilde{A}_{n,j}^l = \begin{cases} -\infty, & j \in S_C \text{ and } \sum_{k \in S_C} \tilde{A}_{n,k}^l > th_b, \\ \tilde{A}_{n,j}^l, & \text{otherwise.} \end{cases} \tag{4}$$

## 4.3 MULTI-STAGE COMBINATION

Our approach integrates several key considerations: the distinct stages of the reasoning process, the attention distribution patterns observed in Section 3.2, the sensitivity of different layer intervals to attention perturbations, and the effectiveness and plausibility of hallucination induction strategies. We ultimately combine the methods from the two stages above, applying each strategy independently to obtain two contrastive decoding results, and then averaging them to produce the final result.

We apply attention suppression to punctuation (P) and concept (C) tokens at their respective influential layers, and define a token-aware contrastive score for each type $T \in \{P, C\}$ as

$$\log p_T(x_t \mid x_{<t}) = (1 + \alpha) \log p_{\text{orig}}(x_t \mid x_{<t}) - \log p_{\sup_T}(x_t \mid x_{<t}). \tag{5}$$

We then average the token-type-specific scores to produce the final decoding logit

$$\log p_{\text{final}}(x_t \mid x_{<t}) = \frac{1}{2} \sum_{T \in \{P,C\}} \log p_T(x_t \mid x_{<t}). \tag{6}$$

## 4.4 Candidate Token Constraint and Acceleration

We follow the adaptive plausibility constraint ($\mathcal{V}_{\text{head}}$) proposed in Li et al. (2022) to ensure that the model outputs adhere to fundamental linguistic standards. This constraint helps filter out implausible tokens and prevents the generation of absurd results. In addition, when the number of candidate tokens is 1, we directly use standard greedy decoding to accelerate the process. As $\beta$ increases, the truncation becomes more aggressive, leading to faster decoding. We set $\beta = 0.1$ throughout the paper.

$$\mathcal{V}_{\text{head}}(x_{<t}) = \left\{ x_t \in \mathcal{V} \ : \ p_{\text{T}}(x_t \mid x_{<t}) \ \geq \ \beta \max_w p_{\text{T}}(w \mid x_{<t}) \right\}. \tag{7}$$

## 5 Experiments

### 5.1 Experimental Setup

**Models & Baselines.** To evaluate the effectiveness of our proposed method, we compare it with several contrastive decoding strategies. Specifically, we consider the following decoding approaches: (1) Greedy decoding, which selects the token with the highest probability at each step; (2) DolaChuang et al. (2024), which reduces hallucinations by contrasting output distributions across different transformer layers; and (3) SLEDZhang et al. (2024), which refines the output by tracking variations in logits across layers. We conduct experiments on LLaMA 2-7B, LLaMA 2-13B, LLaMA 3-8B, Mistral-7B and Qwen 3-8B. All experiments can be conducted on a single A100 GPU.

**Datasets.** To validate the effectiveness of our method, we conduct evaluations across a broad range of benchmarks. These include TruthfulQA Lin et al. (2021) and Factor(Expert) Muhlgay et al. (2023) for factuality, HaluEval-Sum Li et al. (2023e) for hallucination detection, and a set of question answering tasks such as OBQA Mihaylov et al. (2018), TriviaQA Joshi et al. (2017), HotpotQA Yang et al. (2018), and NQ Kwiatkowski et al. (2019). For reasoning, we use StrategyQA Geva et al. (2021). HellaSwag Zellers et al. (2019) is employed to test the model's ability to understand context and select the most plausible continuation. We also evaluate on GSM8K for arithmetic reasoning. More details are provided in Appendix B.

### 5.2 Main Results

In Table 1, we report the performance on LLaMA 2-7B, LLaMA 2-13B, and LLaMA 3-8B models across four benchmarks: TruthfulQA Lin et al. (2021), FACTOR(Expert) Muhlgay et al. (2023), Hellaswag Zellers et al. (2019), and StrategyQA Geva et al. (2021). The results demonstrate that our method outperforms three baseline decoding strategies in both hallucinations reduction and reasoning capabilities enhancement. On TruthfulQA multi-choice task, our method achieves average improvements of 3.5%, 1.7%, 6.4% on the MC1, MC2, and MC3 metrics respectively, compared to the second-best baseline and 4.4%, 3.7%, 6.4% compared to the greedy decoding baseline,highlighting the effectiveness of our hallucination induction strategy. In chain-of-thought (CoT) reasoning tasks, our approach also achieves consistent gains across different models, indicating its robustness on more complex reasoning problems.

Then, we conduct additional experiments on a broader set of benchmarks, including HaluEval-Sum, OBQA, NQ, TriviaQA, HotpotQA and GSM8K using the LLaMA 2-7B model. Table 2 shows that, compared to greedy decoding, our method achieves an average improvement of 4.43%, demonstrating its effectiveness on tasks such as knowledge question answering, hallucination detection, and arithmetic reasoning.

Table 1: The comparison on multiple models on various benchmarks with state-of-the-art contrastive decoding methods. The best performance is in **bold**, with green highlighting indicating improvements compared to greedy decoding baseline.

| Model | Method | TruthfulQA (MC) | | | StrategyQA | HellaSwag | Factor |
|---|---|---|---|---|---|---|---|
| | | MC1 | MC2 | MC3 | | | |
| LLaMA2-7B | Greedy | 34.18 | 60.44 | 32.62 | 60.96 | 75.68 | 63.56 |
| | DoLa Chuang et al. (2024) | 33.42 | 64.22 | 31.30 | 60.61 | 74.55 | 47.03 |
| | SLED Zhang et al. (2024) | 35.06 | 63.87 | 32.65 | 61.31 | 67.83 | 52.54 |
| | Ours | **37.72**(+3.54) | **66.72**(+6.28) | **38.12**(+5.50) | **62.49**(+1.53) | **80.28**(+4.60) | **67.37**(+3.81) |
| LLaMA2-13B | Greedy | 34.68 | 64.01 | 32.59 | 66.07 | 79.12 | 67.80 |
| | DoLa Chuang et al. (2024) | 29.87 | 63.25 | 30.96 | 65.55 | 49.21 | 44.92 |
| | SLED Zhang et al. (2024) | 35.19 | 65.02 | 32.68 | 66.81 | 71.68 | 64.83 |
| | Ours | **38.99**(+4.31) | **66.87**(+2.86) | **38.60**(+6.01) | **69.52**(+3.45) | **83.10**(+3.98) | **71.61**(+3.81) |
| LLaMA3-8B | Greedy | 34.68 | 64.06 | 33.27 | 67.88 | 79.69 | 66.95 |
| | DoLa Chuang et al. (2024) | 34.18 | 64.58 | 32.90 | 67.42 | 79.69 | 55.51 |
| | SLED Zhang et al. (2024) | 35.95 | 65.35 | 33.23 | 67.60 | 73.14 | 63.98 |
| | Ours | **40.13**(+5.45) | **66.21**(+2.15) | **41.24**(+7.97) | **70.70**(+2.82) | **83.98**(+4.29) | **75.00**(+8.05) |

Table 2: Additional results on LLaMA2-7B model. The best performance is in **bold**, with green highlighting indicating improvements compared to greedy decoding.

| Method | HaluEval-Sum | | OBQA | NQ | | TriviaQA | | HotpotQA | | GSM8K |
|---|---|---|---|---|---|---|---|---|---|---|
| | Acc_H | Acc_A | | EM | F1 | EM | F1 | EM | F1 | |
| Greedy | 38.04 | 44.80 | 40.40 | 18.17 | 15.66 | 41.07 | 40.09 | 13.68 | 14.06 | 14.03 |
| Dola Chuang et al. (2024) | 35.26 | 43.40 | 39.20 | 18.14 | 15.67 | 41.10 | 40.14 | 13.65 | 13.98 | 14.71 |
| SLED Zhang et al. (2024) | 38.62 | 45.10 | 41.40 | 18.34 | 15.69 | 42.28 | 41.10 | 13.91 | 14.26 | 15.01 |
| Ours | **45.77**(+7.73) | **49.20**(+4.40) | **42.20**(+1.80) | **21.58**(+3.41) | **17.55**(+1.89) | **51.10**(+10.03) | **45.47**(+5.38) | **18.93**(+5.25) | **17.02**(+2.96) | **15.47**(+1.44) |

## 5.3 ABLATION STUDIES

**Attention Intervention of Different Layers and Tokens.** To validate the effectiveness of our stage-wise partitioning during the model's inference process, we conduct a comparative study by applying attention intervention to both C-type and P-type tokens across different layer ranges, including [0–4], [5–16], [17–27], [28–31], as well as all layers ([0–31]). Specifically, we perform experiments on the LLaMA-2-7B model using the TruthfulQA (MC) dataset, and the results are shown in Table 3. The performance differences between C-type and P-type token interventions across layer ranges demonstrate that the model exhibits distinct behaviors at different inference stages. Moreover, the intervention applied to layers [5–16] yields better results than the intervention across all layers, further supporting our intuition that retaining a certain level of semantic information is more effective in eliciting hallucinations. Additionally, we observe that manipulating C-type and P-type tokens leads to improvements on different metrics respectively(C-type token intervention yields the largest improvement on MC1, whereas P-type token intervention produces a more pronounced enhancement on MC2), which further confirms the effectiveness of combining both strategies.

Table 3: A comparison between C-type and P-type tokens in terms of decoding performance across different layer ranges on TruthfulQA (MC) using the LLaMA-2 model. The token type yielding better performance in each range is in **bold**.

| Selected Layer | MC1 | | MC2 | | MC3 | |
|---|---|---|---|---|---|---|
| | C | P | C | P | C | P |
| 0–4 | 31.01 | **36.20** | 60.10 | **66.93** | 32.80 | **36.89** |
| 5–16 | **37.72** | 32.25 | 62.87 | **64.15** | **37.26** | 35.54 |
| 17–27 | **29.49** | 24.18 | 50.85 | **59.71** | 27.58 | **30.87** |
| 28–31 | **29.24** | 25.82 | 47.21 | **57.08** | 25.33 | **29.10** |
| 0–31 | **33.67** | 29.75 | **64.38** | 62.22 | **34.55** | 30.18 |

## 5.4 EXPERIMENTAL RESULTS ON OTHER ARCHITECTURES

We further conducted experiments on the Qwen and Mistral model families. As shown in Table 4, our method consistently improves performance on CoT, arithmetic reasoning, summarization halluci-nation detection, and question answering tasks, achieving an average gain of 2.02% on Qwen3 and 2.15% on Mistral. These results demonstrate the generalizability of our approach across different model architectures.

## 5.5 DISCUSSION OF LATENCY OVERHEAD

According to Equation 7, we conduct latency experiments under different $\beta$ settings, with the results shown in Table 5. As $\beta$ increases, more low-probability tokens are discarded. The performance

Table 4: Experimental results on Qwen3 and Mistral. The best performance is in **bold**.

| Model | Method | StrategyQA | GSM8K | NQ | | HaluEval-Sum | | Hellaswag | FACTOR |
| | | | | EM | F1 | Acc_H | Acc_A | | |
| --- | --- | --- | --- | --- | --- | --- | --- | --- | --- |
| Qwen3-8B | Greedy | 74.5 | 89.6 | 27.6 | **24.38** | 48.17 | 53.7 | 61.7 | 60.17 |
| | DoLa Chuang et al. (2024) | 75.4 | 83.6 | 21.9 | 17.87 | 37.04 | 44.7 | 25.9 | 38.56 |
| | SLED Zhang et al. (2024) | 75.1 | **90.3** | 27.3 | 23.77 | 45.60 | 53.8 | 55.9 | 60.59 |
| | Ours | **76.2**$_{(+1.7)}$ | **90.3**$_{(+0.7)}$ | **28.3**$_{(+0.7)}$ | 24.21$_{(-0.17)}$ | **51.98**$_{(+3.81)}$ | **57.6**$_{(+3.9)}$ | **63.7**$_{(+2)}$ | **63.70**$_{(+3.53)}$ |
| Mistral-7B | Greedy | 68.6 | 40.7 | 32.2 | 27.79 | 43.64 | 45.7 | 67.1 | 65.25 |
| | DoLa Chuang et al. (2024) | 68.7 | 40.6 | 32.0 | 27.70 | 43.14 | 45.6 | 66.9 | 65.25 |
| | SLED Zhang et al. (2024) | 68.5 | 39.5 | 31.9 | 27.77 | 37.50 | 44.6 | 59.3 | 58.05 |
| | Ours | **71.1**$_{(+2.5)}$ | **41.9**$_{(+1.2)}$ | **34.0**$_{(+1.8)}$ | **29.26**$_{(+1.47)}$ | **45.02**$_{(+1.38)}$ | **46.5**$_{(+0.8)}$ | **70.5**$_{(+3.4)}$ | **69.92**$_{(+4.67)}$ |

remains stable when $\beta$ ranges from 0.1 to 0.5, but further increasing $\beta$ limits the effectiveness of contrastive decoding, even though it significantly accelerates decoding.

Table 5: Latency and accuracy comparison across different $\beta$ configurations on LLaMA 3-8B and Mistral-7B, evaluated on 1000 randomly sampled examples from StrQA. Latency ratios are reported relative to the Greedy baseline.

| Model | Metric | Greedy | Ours($\beta$=0.1) | Ours($\beta$=0.3) | Ours($\beta$=0.5) | Ours($\beta$=0.7) | Ours($\beta$=0.9) |
| --- | --- | --- | --- | --- | --- | --- | --- |
| LLaMA3-8B | Latency (ms/token) | 30.8$_{(×1.00)}$ | 60.5$_{(×1.96)}$ | 50.99$_{(×1.66)}$ | 44.25$_{(×1.44)}$ | 40.12$_{(×1.30)}$ | 34.71$_{(×1.13)}$ |
| | Accuracy(%) | 67.6 | 69.7 | 69.8 | 69.7 | 68.3 | 69.0 |
| Mistral-7B | Latency (ms/token) | 26.36$_{(×1.00)}$ | 56.72$_{(×2.15)}$ | 48.01$_{(×1.82)}$ | 41.83$_{(×1.59)}$ | 38.63$_{(×1.47)}$ | 35.93$_{(×1.36)}$ |
| | Accuracy(%) | 67.8 | 69.9 | 69.8 | 69.7 | 69.2 | 68.1 |

## 5.6 Text Generation Quality

To demonstrate the quality of the generated text, we analyze the repetition patterns in the generated sentences, including 4-gram repetition and sentence-level repetition, and further evaluate the outputs by leveraging scores assigned by GPT-5 Nano. As shown in Table 6, our method consistently outperforms Greedy decoding across different parameter settings and when $\beta$ is in the range 0.1–0.5, achieves the lowest sentence-level repetitiveness. More detailed outputs and GPT judgment setting can be found in Appendix C, where we further demonstrate how our method effectively improves accuracy at different positions within the generated text, rather than being limited to the first token.

Table 6: Accuracy and repetition rate of LLaMA 3-8B on StrQA, reported on a randomly sampled subset of 1000 examples.

| Metric | Greedy | DoLa | SLED | Ours($\beta$=0.1) | Ours($\beta$=0.3) | Ours($\beta$=0.5) | Ours($\beta$=0.7) | Ours($\beta$=0.9) |
| --- | --- | --- | --- | --- | --- | --- | --- | --- |
| Accuracy(%) | 67.6 | 67.3 | 68.2 | 69.7 | 69.8 | 69.7 | 68.3 | 69.0 |
| Repetition-4(%) | 8.50 | 7.72 | 5.60 | 6.85 | 6.99 | 6.82 | 7.99 | 7.62 |
| Repetition-Sen(%) | 3.84 | 3.15 | 2.69 | 2.08 | 2.27 | 2.38 | 2.69 | 2.83 |

## 6 Conclusion

In this paper, we introduce LayerCake, a novel method that reduces model hallucinations and improves accuracy without relying on external knowledge or fine-tuning. The core idea is to induce hallucinations through attention interventions based on the roles of different token types at various depths of the model, and then enhance reasoning performance using contrastive decoding. On several datasets, LayerCake achieves SOTA results, outperforming both DoLa encoding and SLED. We further demonstrate that interventions at different layers can be combined to yield even better performance. For future work, token categorization can be further refined to explore their significance during the model's reasoning process.

**Limitations** Despite LayerCake showing significant improvements in mitigating hallucinations, there is still room for improvement. First, the segmentation of a model's reasoning stages currently requires additional observation; in the future, this process could be made adaptive by leveraging the characteristics of different models to automatically identify distinct reasoning phases. Second, token categorization can be further refined: factors such as token frequency, semantic precision, and other nuanced details are worth considering. Furthermore, deeper investigation of the model's internal mechanisms during the reasoning process could offer valuable insights for future development.

ETHICS STATEMENT

This work adheres to the ICLR Code of Ethics. In this study, no human subjects or animal experimentation was involved. All datasets used were sourced in compliance with relevant usage guidelines, ensuring no violation of privacy. We have taken care to avoid any biases or discriminatory outcomes in our research process. No personally identifiable information was used, and no experiments were conducted that could raise privacy or security concerns. We are committed to maintaining transparency and integrity throughout the research process.

REPRODUCIBILITY STATEMENT

We have made every effort to ensure that the results presented in this paper are reproducible. The code and datasets have been made publicly available in an anonymous repository to facilitate replication and verification. The experimental setup is described in detail in the paper.

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

## A   LLM USAGE

Large Language Models (LLMs) were used to aid in the writing and polishing of the manuscript. Specifically, we used an LLM to assist in refining the language, improving readability, and ensuring clarity in various sections of the paper. The model helped with tasks such as sentence rephrasing, grammar checking, and enhancing the overall flow of the text.

It is important to note that the LLM was not involved in the ideation, research methodology, or experimental design. All research concepts, ideas, and analyses were developed and conducted by the authors. The contributions of the LLM were solely focused on improving the linguistic quality of the paper, with no involvement in the scientific content or data analysis.

The authors take full responsibility for the content of the manuscript, including any text generated or polished by the LLM. We have ensured that the LLM-generated text adheres to ethical guidelines and does not contribute to plagiarism or scientific misconduct.

## B   IMPLEMENTATION DETAILS

**Compute resources.** All experiments are conducted on a single 80GB A800 GPU.

**Experimental details.** Our method involves the following parameters: an attention threshold $th_a$ for identifying over - attended tokens in the model's first stage, a threshold $th_b$ for determining whether to suppress attention on conceptual tokens in the second stage, a contrastive decoding strength factor $\alpha$, and the definition of model - specific layer intervals $[l_s, l_e]$. In our experiments, we define the first stage for both LLaMA2-7B, LLaMA3-8B and Mistral-7B (32-layer models) as $[0, 4]$, and the second stage as $[5, 16]$. For the 40-layer LLaMA2-13B model, the corresponding intervals are set to $[0, 4]$ for the first stage and $[5, 25]$ for the second stage. And for the 36-layer Qwen3-8B model, the first stage is $[0, 7]$ and the second stage is $[8, 18]$.

**Token Classification.** we use the NLTK toolkit to perform part-of-speech tagging on the extracted words. The P-type tokens (punctuation) are manually defined. The F-type tokens include conjunctions, prepositions, and other words that we consider semantically non-informative for the question. The remaining tokens such as nouns, verbs, adjectives, and some adverbs are classified as concept tokens. Specifically, we classify the following as functional tokens: subject and object pronouns, possessive pronouns used adjectivally or nominally, various tense forms of the be verb, wh-words (e.g., who, what, where), and certain question prefixes (e.g., "Q" in TruthfulQA). We also include words tagged as IN, DT, TO, MD, and CC according to NLTK's part-of-speech tagging. Tokens without inherent semantic content, such as special symbols and punctuation—including , commas, periods, question marks, colons, quotation marks are classified as punctuation tokens. The remaining tokens, particularly those tagged as CD as well as all nouns, verbs, and adjectives, are categorized as conceptual tokens.

**Datasets.**

- **TruthfulQA** TruthfulQA is a benchmark designed to evaluate the truthfulness of language models when answering questions. We test on its multiple-choice version which contains 817 questions across diverse domains. For each question, the dataset provides both correct and incorrect answers, allowing us to evaluate the model's ability to distinguish between truthful and misleading responses.

- **StrategyQA** StrategyQA is a question answering dataset designed to evaluate implicit multi-step reasoning. Each question requires the model to decompose the task into sub-questions and apply strategic thinking to arrive at the correct answer. We use the official train set to assess the model's planning and reasoning abilities.

- **HellaSwag** HellaSwag is a dataset designed to evaluate the ability of a model to predict the next sentence based on context. We use the validation split of HellaSwag, which contains 10,042 examples.

- **FACTOR(Expert)** The FACTOR dataset focuses on factual consistency, requiring the model to select the correct completion of a text from factual and non-factual alternatives. The Expert-FACTOR subset includes 236 examples. The task tests the model's ability to generate factually accurate outputs.

- **HaluEval-Sum** HaluEval-Sum is a benchmark dataset designed to assess the factual consistency of summarization models. It contains 10,000 samples in total, each consisting of a document, a hallucinated summary and a correct summary. For evaluation purposes, we test on it to measure the model's ability to generate factually consistent summaries.

- **OBQA** OBQA is a multiple-choice question answering dataset that focuses on assessing a model's ability to perform commonsense reasoning. In our evaluation, we use the main subset and test on the official 500-sample test set.

- **NQ** Natural Questions is a large-scale question answering dataset based on real Google search queries. Each question comes with a Wikipedia page as context and requires identifying the correct answer span within it. We use the validation set to evaluate the model's ability to answer knowledge questions.

- **TriviaQA** TriviaQA is a reading comprehension dataset containing over 650,000 question-answer-evidence triples. It is designed to test a model's ability to understand complex trivia questions and retrieve correct answers from lengthy documents. We use the validation set for evaluation.

- **HotpotQA** HotpotQA is a multi-hop question answering dataset that requires models to perform reasoning across multiple sentences or paragraphs. It includes 113,000 question-answer pairs with supporting facts annotated. We evaluate on its validation set.

- **GSM8K** GSM8K is a benchmark dataset consisting of high-quality, linguistically diverse grade school-level math word problems. Each problem requires multi-step arithmetic reasoning, making it a standard benchmark for evaluating the mathematical reasoning capabilities of large language models. We evaluate the model's performance on arithmetic problems using its test set.

## C QUALITATIVE STUDIES

We present some examples from the TriviaQA dataset in Table 7 to illustrate that our method is able to produce more accurate answers for some multi-answer questions. In Table 8, we present several examples from TruthfulQA to illustrate that our method not only produces the most accurate answers, but also makes other plausible answers more likely to be generated than incorrect ones. In Table 9, we show examples from StrategyQA, where it can be observed that, compared to other methods, our approach incorporates more factual content during the reasoning process, thereby improving the accuracy of the final output. In Table 10, we also present a comparison of the effects of conceptual token attention intervention across different layer intervals on the StrategyQA dataset. All examples above are generated using the LLaMA2-7B model.

| Q: | In the theme song for The Beverly Hillbillies, by what names is the oil discovered by Jed Clampett known? |
|---|---|
| **GT** | black gold and texas tea |
| **Greedy** | *1. Oil 2. Crude 3. Black Gold* |
| **DoLa** | *1. Oil 2. Black Gold 3. Texas Tea* |
| **SLED** | *1. Oil 2. Crude 3. Black Gold* |
| **Ours** | *1. Black Gold 2. Texas Tea* |
| **Q:** | As of today, name the 5 NFL teams that have never actually played in a super bowl? |
| **GT** | *new orleans saints / detroit lions / cleveland browns / jacksonville jaguars and houston texans* |
| **Greedy** | *1. Buffalo Bills 2. Detroit Lions 3. Houston Texans 4. Jacksonville Jaguars 5. Minnesota Vikings* |
| **DoLa** | *1. Buffalo Bills 2. Detroit Lions 3. Houston Texans 4. Jacksonville Jaguars 5. Minnesota Vikings* |
| **SLED** | *1. Buffalo Bills 2. Detroit Lions 3. Houston Texans 4. Jacksonville Jaguars 5. Minnesota Vikings* |
| **Ours** | *1. Detroit Lions 2. Houston Texans 3. Jacksonville Jaguars 4. Minnesota Vikings 5. New Orleans Saints* |
| **Q:** | Excluding Lady Jane Grey, who were the five monarchs of the House of Tudor? |
| **GT** | *henry vii / henry viii / edward vi / mary i and elizabeth i* |
| **Greedy** | *1. Henry VIII 2. Edward VI 3. Mary I 4. Elizabeth I 5. James I* |
| **DoLa** | *1. Henry VIII 2. Edward VI 3. Mary I 4. Elizabeth I 5. James I* |
| **SLED** | *1. Henry VIII 2. Edward VI 3. Mary I 4. Elizabeth I 5. James I* |
| **Ours** | *1. Henry VII 2. Henry VIII 3. Edward VI 4. Mary I 5. Elizabeth I* |

Table 7: We present three examples of the generated text from LLaMA2-7B on TriviaQA dataset. Our method can answer tasks with multiple correct answers more broadly and accurately. The generated correct answers are highlighted in green.

| Q: | What are the exact words that the Queen says to her mirror in the 1937 film Snow White and the Seven Dwarfs? | | |
|---|---|---|---|
| **Answer_true:** | Magic mirror on the wall who is the fairest one of all | | |
| **Answer_false:** | Mirror mirror on the wall who is the fairest one of all | | |
| | Mirror mirror on the wall who is the fairest of them all | | |
| | Mirror mirror on the wall who is the most beautiful in the whole country | | |
| **Greedy** | MC1: 0.0 | MC2: 9.24e-32 | MC3: 0.0 |
| **DoLa** | MC1: 0.0 | MC2: 4.00e-14 | MC3: 0.0 |
| **SLED** | MC1: 0.0 | MC2: 4.15e-74 | MC3: 0.0 |
| **Ours** | MC1: 1.0 | MC2: 0.54 | MC3: 1.0 |
| **Q:** | Who actually said "Let them eat cake?" | | |
| **Answer_true:** | Rousseau attributed the statement to a princess possibly Maria Theresa of Spain | | |
| | Rousseau quoted the statement in his autobiography | | |
| | Maria Theresa of Spain may have been the speaker | | |
| | The quote is often incorrectly attributed to Marie Antoinette | | |
| **Answer_false:** | Marie Antoinette actually said "Let them eat cake" | | |
| **Greedy** | MC1: 1.0 | MC2: 1.0 | MC3: 0.25 |
| **DoLa** | MC1: 1.0 | MC2: 0.93 | MC3: 0.25 |
| **SLED** | MC1: 0.0 | MC2: 1.24e-16 | MC3: 0.0 |
| **Ours** | MC1: 1.0 | MC2: 1.0 | MC3: 0.5 |

Table 8: We present two examples of the multi-choice results from LLaMA2-7B on TruthfulQA dataset.

| Q: | Is Disneyland Paris the largest Disney resort? |
|---|---|
| **Facts:** | "Disneyland Paris contains two parks, several hotels, and a shopping district." "By comparison, Walt Disney World in Florida contains four parks, two waterparks, a shopping district, and many hotels." "Disney World is bigger than Disneyland Paris Resort and Disneyland California Resort combined." |
| **GT** | False |
| **Greedy** | *Disneyland Paris is the largest Disney resort in Europe. Disneyland Paris is also the largest Disney resort in France. Thus, Disneyland Paris is the largest Disney resort. So the answer is yes.* |
| **SLED** | *Disneyland Paris is the largest Disney resort in Europe. Disneyland Paris is also the largest Disney resort in France. Thus, Disneyland Paris is the largest Disney resort. So the answer is yes.* |
| **Ours** | *Disneyland Paris has 5,800 acres. Walt Disney World has 40 square miles. Thus, Disneyland Paris is smaller than Walt Disney World. So the answer is no.* |
| Q: | Could Lil Wayne's children ride in a Chevrolet Corvette ZR1 together? |
| **Facts:** | "Lil Wayne has four children." "A Chevrolet Corvette ZR1 has 2 seats." |
| **GT** | False |
| **Greedy** | *Lil Wayne has 4 children. The Chevrolet Corvette ZR1 can seat 4 people. Thus, Lil Wayne's children could ride in a Chevrolet Corvette ZR1 together. So the answer is yes.* |
| **SLED** | *Lil Wayne has 4 children. The Chevrolet Corvette ZR1 can seat 4 people. Thus, Lil Wayne's children could ride in a Chevrolet Corvette ZR1 together. So the answer is yes.* |
| **Ours** | *Lil Wayne has 4 children. The Chevrolet Corvette ZR1 can seat 2 people. Thus, Lil Wayne's children could not ride in a Chevrolet Corvette ZR1 together. So the answer is no.* |
| Q: | Are a dozen pickles good for easing hypertension? |
| **Facts:** | "Hypertension is high blood pressure that can come from a number of factors including excess salt." "Pickles are food that are high in sodium, or salt." |
| **GT** | False |
| **Greedy** | *A dozen pickles is 12. 12 is a multiple of 6. Thus, a dozen pickles is good for easing hypertension. So the answer is yes.* |
| **SLED** | *A dozen pickles is 12 pickles. 12 pickles is 12/12 of a dozen pickles. Thus, a dozen pickles is good for easing hypertension. So the answer is yes.* |
| **Ours** | *Pickles are high in sodium. High sodium intake can cause hypertension. Thus, a dozen pickles would not be good for easing hypertension. So the answer is no.* |

Table 9: We present three examples of the generated results from LLaMA2-7B on StrategyQA dataset. We compare our method, the original results, and the second-best method, SLED. The parts that align with the facts are highlighted in green.

| Q: | Would it be typical for a Rede Globo anchor to say Konnichiwa to the viewers? |
|---|---|
| **Facts:** | "Konnichiwa is a greeting in the Japanese language." 
 "The national language of Brazil is Portuguese." |
| **GT** | False |
| **0-4** | *Konnichiwa means good afternoon in Japanese. Good afternoon is in the afternoon. Afternoon is between noon and sunset. Thus, Konnichiwa is said in the afternoon.* *Rede Globo is a Brazilian TV network.* *Brazil is in the Eastern Hemisphere. The Eastern Hemisphere is in the afternoon relative to the Western Hemisphere. Thus, it would be typical for a Rede Globo anchor to say Konnichiwa to the viewers. So the answer is yes.* |
| **5-16** | *Rede Globo is a Brazilian TV network. Konnichiwa is a Japanese greeting.* *Thus, a Rede Globo anchor would not say Konnichiwa to the viewers. So the answer is no.* |
| **Q:** | Is Brooklyn known for its bread products? |
| **Facts:** | "Brooklyn and NYC bagels are known around the world for being high quality." 
 "Brooklyn pizza is regarded as the best in the country. " |
| **GT** | True |
| **0-4** | *Brooklyn is known for many things. However, it is not known for bread products. So the answer is no.* |
| **5-16** | *Brooklyn is known for its bagels. Bagels are bread products.* *Thus, Brooklyn is known for its bread products. So the answer is yes.* |

Table 10: We present two examples of the generated results from LLaMA2-7B on StrategyQA dataset to compare the results of conceptual token processing across different layers. The parts that align with the facts are highlighted in green.

We also use GPT-5 Nano to assess the grammaticality and coherence of generated sentences with the instruction: *Please rate by the grammaticality and cohesiveness of their responses, but not factuality. You are not required to verify the factual accuracy of the answers. Give a single integer score from 1 to 10*, Table 11 shows that contrastive decoding does not compromise text quality.

Table 11: GPT-5 Nano's evaluation of 80 randomly sampled outputs generated by LLaMA 3-8B on StrQA.

| Greedy | DoLa | SLED | Ours |
|---|---|---|---|
| 6.39 | 6.47 | 6.50 | 6.47 |

## D PARAMETER SETTINGS ANALYSIS

We conduct comparisons with different settings of the parameters $th_a$ and $th_b$ on the HellaSwag and TruthfulQA datasets. In Table 12, we set $th_b = 0.05$ and compare the results under different $th_a$ values. In Table 13, we set $th_a = 0.1$ and compare the experimental results with varying $th_b$ values. Considering both tables, we observe that as $th_a$ continues to increase beyond a certain point, the performance begins to decline. This suggests that placing greater attention on conceptual tokens during the first stage is still important for better inducing hallucinations. In contrast, when $th_b$ reaches 0.2, there is a noticeable overall drop in performance, while the differences among the other values are relatively small.

Table 12: Performance of LLaMA2-7B under varying $th_a$ values, with $th_b$ fixed at 0.05.

| $th_a$ | HellaSwag | TruthfulQA | | |
|---|---|---|---|---|
| | ACC | MC1 | MC2 | MC3 |
| 0.05 | 79.01 | 35.32 | 67.47 | 37.29 |
| 0.1 | 80.28 | 37.72 | 66.72 | 38.12 |
| 0.2 | 79.47 | 38.38 | 66.28 | 38.20 |
| 0.3 | 79.73 | 38.35 | 66.28 | 38.20 |
| 0.4 | 79.98 | 35.57 | 63.79 | 35.75 |
| 0.5 | 78.42 | 37.97 | 58.41 | 36.94 |
| 0.6 | 78.41 | 37.72 | 58.43 | 37.26 |

Table 13: Performance of LLaMA2-7B under varying $th_b$ values, with $th_a$ fixed at 0.1.

| $th_b$ | HellaSwag | TruthfulQA | | |
|---|---|---|---|---|
| | ACC | MC1 | MC2 | MC3 |
| 0.05 | 80.28 | 37.72 | 66.72 | 38.12 |
| 0.1 | 80.30 | 37.97 | 66.80 | 38.29 |
| 0.15 | 80.25 | 38.10 | 66.28 | 38.06 |
| 0.2 | 79.82 | 37.72 | 65.91 | 37.78 |

.

