# OpenReview forum: "LayerCake: Token-Aware Contrastive Decoding within Large Language Model Layers"
_ICLR.cc/2026/Conference — Submitted to ICLR 2026_

### Official Review · Reviewer_RPM2 · 2025-10-20

**Soundness:** 2
**Presentation:** 3
**Contribution:** 2
**Rating:** 4
**Confidence:** 4

**Summary:**

The paper introduces a new contrastive decoding technique (named "LayerCake") to improve the factuality of LLM generations. Extensive evaluations on benchmarks using various models (llama, mistral, qwen) show promising results of the technique with reasonable tradeoffs in the inference latency.

**Strengths:**

The experiments and evaluations are comprehensive and clearly convey the message of the paper. The writing and presentation of the methodology is also clear.

**Weaknesses:**

The method relies heavily on token type characterizations, additional hyper-parameters such as $\alpha, th_a, th_b, \beta$ and determining layers for stage 1,2 of interventions that are model specific. Since the authors have presented ablations but not guidance on how to select these values, (especially $\alpha$) in Eq(5), employing this technique can be cumbersome for unknown datasets or new models.

More importantly, this presents a situation where one has to tune a lot of hyper-parameters to obtain LLM outputs and verify if they are correct or not based on established benchmarks/verifiers. It would be beneficial if the authors discuss situations where verifiers are not available.

**Questions:**

1. What are the hyper-parameter values of $\alpha, th_a, th_b$ used for the results in Tables 1-6? There is an ablation in the Appendix but I couldn't find the $\alpha$ values. More importantly, is there a guidance for choosing them?

2. Regarding the comparison with SLED, I noticed that the Gemma-1B model performance reported in their paper is quite high (when compared to llama-3-8B) on TruthfulQA. Is it possible to have such an apples-apples comparison in this paper so that the readers have a clear picture of the effectiveness of LayerCake? I am asking because in the current paper, the authors show that the performance of Llama models on FACTOR with DOLA and SLED in Table 1 is lower than greedy, whereas SLED was shown to have better performance than greedy decoding for various gemma models. Clearing this confusion can be beneficial for future researchers.

3. For inference on an unknown dataset with long input sequences, how should one go about selecting the hyper-parameters? since the latency depends on $\beta$, what can be a guiding principle for choosing the optimal value?

4. Please fix the format of references (usage of "\ cite{}") throughout the text.

5. Figure 3(b) label: rename vallina -> vanilla

---

> ### Author Response · Authors · 2025-11-22
> **Response to Reviewer RPM2 (1/2)**
>
> Dear Reviewer RPM2,
>
> Thank you for your valuable suggestions and questions. Here is our response:
>
> **Q1:Choice of Hyperparameters**
>
> Our work involves several hyperparameters, including $α$, $th_a$, $th_b$, $β$, and the layer ranges. These parameters fall into two categories. Among them, $th_a$, $th_b$ and the layer ranges are closely related to the intrinsic characteristics of the model. In our experiments, they are determined based on the attention distribution patterns of the model. The remaining parameters, $α$ and $β$, are mainly associated with contrastive decoding.
>
> We analyzed attention weights of different models on tasks with varying input lengths. The table below reports the top-3 attention values of P-type tokens on the 2nd and 10th layers—representative mid-layers of the first and second stages—as well as the summed C-type attention for LLaMA2-7B, LLaMA3-8B, and Mistral-7B on representative cases from StrategyQA, NQ, and HaluEval-Sum. Across inputs of 64 (NQ), 438 (StrategyQA), and 1672 (HaluEval-Sum) tokens, P-type tokens consistently show a sharp drop between the high-weight tokens and the rest. Based on this, we set $th_a$=0.1 for computational convenience and keep it fixed in all experiments.
>
> **Layer 2**
>
> | Model   | Type    | StrategyQA               | HaluEval-Sum            | NQ                     |
> |---------|---------|---------------------------|--------------------------|-------------------------|
> | llama2  | P top3  | 0.462 / 0.350 / 0.019     | 0.534 / 0.315 / 0.012    | 0.473 / 0.407 / 0.020   |
> |         | C sum   | 0.037                     | 0.044                    | 0.028                   |
> | llama3  | P top3  | 0.817 / 0.022 / 0.016     | 0.804 / 0.005 / 0.001    | 0.897 / 0.020 / 0.004   |
> |         | C sum   | 0.043                     | 0.135                    | 0.045                   |
> | mistral | P top3  | 0.580 / 0.220 / 0.009     | 0.552 / 0.242 / 0.009    | 0.650 / 0.235 / 0.010   |
> |         | C sum   | 0.069                     | 0.072                    | 0.044                   |
>
> **Layer 10**
> | Model   | Type    | StrategyQA               | HaluEval-Sum            | NQ                     |
> |---------|---------|---------------------------|--------------------------|-------------------------|
> | llama2  | P top3  | 0.262 / 0.220 / 0.053     | 0.205 / 0.197 / 0.053    | 0.390 / 0.292 / 0.038       |
> |         | C sum   | 0.092                     | 0.155                    | 0.175                   |
> | llama3  | P top3  | 0.358 / 0.090 / 0.068     | 0.238 / 0.045 / 0.037    | 0.670 / 0.012 / 0.005      |
> |         | C sum   | 0.133                     | 0.505                    | 0.270                   |
> | mistral | P top3  | 0.324 / 0.103 / 0.066     | 0.325 / 0.092 / 0.063    | 0.390 / 0.111 / 0.063       |
> |         | C sum   | 0.093                     | 0.171                    | 0.279                   |
>
> For $th_b$, we set it to 0.05 in Table 1, while in the subsequent experiments it is set to 0. In fact, we found that this parameter has little practical impact. We will remove it in the revised version and update the corresponding results in Table 1 using $th_b$ = 0. Table 1 will be updated as:
>
> | Model       | Method | TQA MC1 | TQA MC2 | TQA MC3 | TQA-v2 MC1 | TQA-v2 MC2 | TQA-v2 MC3 | StrategyQA| HellaSwag| Factor |
> |-------------|--------|---------|---------|---------|-------------|-------------|-------------|-------|-------|--------|
> | LLaMA2-7B   | Greedy | 34.18   | 60.44   | 32.62   | 26.71       | 41.88       | 19.15       | 60.96 | 75.68 | 63.56  |
> |             | DoLa   | 33.42   | 64.22   | 31.30   | 27.22       | 42.54       | 19.28       | 60.61 | 74.55 | 47.03  |
> |             | SLED   | 35.06   | 63.87   | 32.65   | 23.67       | 47.24       | 22.78       | 61.31 | 67.83 | 52.54  |
> |             | Ours   | 37.72   | 66.74   | 38.14   | 28.73       | 45.85       | 21.13       | 62.66 | 79.09 | 67.80  |
> |  LLaMA2-13B  | Greedy | 34.68   | 64.01   | 32.59   | 27.72       | 43.14       | 19.64       | 66.07 | 79.12 | 67.80  |
> |             | DoLa   | 29.87   | 63.25   | 30.96   | 29.87       | 48.85       | 24.35       | 65.55 | 49.21 | 44.92  |
> |             | SLED   | 35.19   | 65.02   | 32.68   | 24.18       | 46.98       | 23.35       | 66.81 | 71.68 | 64.83  |
> |             | Ours   | 39.62   | 65.98   | 38.73   | 30.89       | 46.70       | 22.06       | 68.38 | 81.86 | 70.34  |
> | LLaMA3-8B   | Greedy | 34.68   | 64.06   | 33.27   | 29.75       | 48.51       | 22.92       | 67.88 | 79.69 | 66.95  |
> |             | DoLa   | 34.18   | 64.58   | 32.90   | 29.75       | 48.45       | 22.97       | 67.42 | 79.69 | 55.51  |
> |             | SLED   | 35.95   | 65.35   | 33.23   | 26.20       | 50.09       | 24.72       | 67.60 | 73.14 | 63.98  |
> |             | Ours   | 37.97   | 67.30   | 39.99   | 33.67       | 52.40       | 25.24       | 68.25 | 82.97 | 73.73  |

---

> ### Author Response · Authors · 2025-11-22
> **Response to Reviewer RPM2 (2/2)**
>
> The choice of layer ranges can be referenced from Figure 3(a) in the paper. We use the growth trend of the summed C - type attention as the key factor in determining this range. These parameters remain consistent across models and different tasks.
>
> For $β$, we follow standard practices in contrastive decoding and fix it to 0.1. The remaining parameter $α$ is adjustable across different tasks. In all experiments conducted after updating Table 1, we set α = 1 by default for all tasks except Factor and TruthfulQA. For TruthfulQA, since our implementation follows the publicly released code of DoLa and SLED, we set $α$ = 0 to remain consistent with their setup. In addition, we conducted supplementary experiments following the updated version of DoLa, where $α$ is set to 1. The resulting performance is shown in updated Table 1, marked as TruthfulQA-v2.
>
> These findings indicate that $α$ = 1 is generally suitable for most tasks, with the exception of Factor, where we set $α$ = 0.5.
> We also conducted an ablation study on $α$, and the results are presented below:
>
> | Model     | α   | TQA-v2 MC1 | TQA-v2 MC2 | TQA-v2 MC3 | StrategyQA| NQ EM | NQ F1 | HotpotQA EM | HotpotQA F1 | Factor |
> |-----------|-----|-------------|-------------|-------------|--------|-------|-------|--------------|-------------|--------|
> | llama2-7b | 0.5 | 30.76       | 51.86       | 25.70       | 63.93 | 23.75 | 18.75 | 19.30        | 17.34       | 67.80  |
> |           | 1   | 28.73       | 45.85       | 21.13       | 62.66 | 21.58 | 17.55 | 18.93        | 17.02       | 62.71  |
> |           | 1.5 | 27.22       | 42.66       | 18.91       | 62.66 | 20.05 | 16.37 | 18.35        | 17.47       | 60.59  |
>
> We observe that $α$ = 0.5 and $α$ = 1 work well for most tasks. Although our method involves several hyperparameters, in practice they do not require careful per - task tuning, as the stable performance across these $α$ values demonstrates.
>
> Regarding the choice of $𝛽$, based on the results in Table 5, we observe that performance remains relatively stable when $β$ is within the range of 0.1–0.5. If inference latency is not a major concern, we typically set β = 0.1, which is also a commonly used choice in contrastive decoding.
>
> **Q2:Comparison with SLED**
>
> In the earlier version of the SLED paper, experimental results for Llama2-7B, Llama2-13B, and Llama3-8B were reported. Using the released SLED code, we were able to reproduce these results and conduct a direct comparison, which we believe is a fair evaluation. However, we found that its performance on the EXPERT-FACTOR is suboptimal. We attempted to tune several parameters, but the performance could not be improved to a satisfactory level.
>
> Thank you for pointing out all the detailed issues in our paper. We will address and correct them one by one in the subsequent revision.

---

> ### Comment · Reviewer_RPM2 · 2025-11-27
>
> Thanks for the response.
>
> Q1. Why is the ablation with different $\alpha$ mentioned above for llama-2-7b not consistent with the updated Table 1 in the paper? For example: which $\alpha$ gives a value of 37.72 on TQA-MC1 with llama-2-7b? Shouldn't this be the baseline to ablate against? Also, I do not think it is fair to choose $\alpha$ as per task performance and then report them under a single table. This gives the wrong impression to the readers.
>
> Q2: More importantly, when the authors state different token types in the paper, what exactly is the distribution of P,F and C tokens that they are suppressing per layer? They simply state that these tokens are computed based on NLTK toolkit but what exactly is the distribution per task? Do these distributions affect the selection of hyperparameters? These key insights are missing.
>
> Q3: Since the authors state that their motivation to suppress tokens is based on the patterns in Figure 3, why can't we select a certain fraction of tokens (say 10%) that have the maximum attention values per layer and suppress them? What is the benefit of token-level classification in these cases? The "attention sinks" in the early layers already indicate the behaviour observed in Figure b, so suppressing tokens in the beginning of the prompt and noticing a drop in accuracy is not surprising. Therefore, I don't see a benefit of token classification if attention values themselves can indicate which tokens are important and which ones are not.

---

> > ### Author Response · Authors · 2025-12-03
> > **Response to Reviewer RPM2**
> >
> > Dear Reviewer RPM2,
> >
> > Thank you for your response. Here is our response:
> >
> > **Q1:Parameter Explanation**
> >
> > The ablation results for $α$ were conducted on the updated version of TQA, namely TQA-v2, whereas the value of 37.72 on TQA-MC1 corresponds to $α$ = 0. To unify the choice of $α$, we updated the results to demonstrate that a single set of parameters can still be applied across most tasks.
> >
> > **Q2:Distribution of Different Tokens**
> >
> > We computed the average distribution of P- and C-type tokens across the three tasks, as shown in the table:
> >
> > | Token Type | HellaSwag | StrategyQA | GSM8K   |
> > |------------|-----------|------------|---------|
> > | C-type     | 0.5401    | 0.5614     | 0.5811  |
> > | P-type     | 0.1078    | 0.2179     | 0.2579  |
> >
> > For each case, the distribution across layers remains consistent. Despite these distributional differences shown in the table, we kept the hyperparameter choices consistent across these tasks to demonstrate the generality of our approach.
> >
> > **Q3:Significance of Token Categorization**
> >
> > Using the same layer selections, we conducted the experiments on Llama2-7B, where we directly intervened on the top 5%, 10%, 20%, and 50% of tokens ranked by attention values and performed contrastive decoding. The results are shown in the table:
> >
> > | Top-k Tokens | StrategyQA | GSM8K | HellaSwag  |
> > |--------------|------------|-------|------------|
> > | 5%           | 60.87      | 12.28 | 67.12      |
> > | 10%          | 62.27      | 12.21 | 74.34      |
> > | 20%          | 61.40      | 12.74 | 74.80      |
> > | 50%          | 61.40      | 12.74 | 75.14      |
> >
> > From the results, directly intervening on tokens with relatively high attention contributions leads to unstable performance, and in some cases—such as GSM8K—performs even worse than the greedy decoding. Our understanding is that, given these clear differences, it is natural to design interventions that leverage the characteristic variation of each token category. This observation is what motivated our series of targeted intervention strategies.

---

### Official Review · Reviewer_cria · 2025-10-26

**Soundness:** 2
**Presentation:** 3
**Contribution:** 2
**Rating:** 4
**Confidence:** 3

**Summary:**

This work proposes a variant of (intra-model) contrastive decoding, where the final next-token distribution is derived via contrast between the vanilla model output logits and the output logits of a lower-quality/perturbed model generation setting. In the method proposed here the low-quality token distribution is created by manipulating the attention within specific model layers, and this is done differentially for different categories of tokens (punctuation, concepts/content words, function words). The authors show some motivating analyses of attention behavior for different token types across different layers, and present experimental results that show how their decoding method leads to improved results for several models over multiple benchmarks, focusing on factuality, hallucinations, and QA.

**Strengths:**

* The general approach makes sense, that it is possible to identify more precise sources of "problematic" LLM behaviors, and use those to perform contrastive decoding that is more informed and targeted towards these areas (rather than a generic "weak model" vs. "expert model" scenario).
* The specific method of using attention interventions on specific token categories is novel as far as I'm aware.
* The results are presented over a wide array of benchmarks for different types of tasks, and show consistent and substantial gains across tasks and models.

**Weaknesses:**

1. The method relies on quite a lot of parameters and heuristics - I was not entirely convinced by the motivation for those, and at the same time the paper is somewhat unclear regarding the methodology for choosing them in practice. Specifically, there is $th_a$ and $th_b$, and $\alpha$; the choice of layers representing the "early" and "middle" stages of processing; a separate attention modification logic for punctuation and conceptual tokens; and the decision to perform contrast with each of those independently and then average the logits. Most of these decisions can be explained to some extent, but there is not really an overarching logic and we end up with a method that is quite complex, does not directly correspond to the analyses in the paper, and may require quite a bit of hyper-parameter optimization to work well (see Q1 below). Regarding the intuitions, several of the motivating statements are a bit vague and not sufficiently backed up by experiments/citations, like "such interventions can disrupt the model’s internal processes at different stages and thus serve as a mechanism to induce hallucinations" (l. 278), "C-type tokens are more likely to trigger hallucination issues" (l. 300), or "…enabling us to adaptively identify layers where the model begins focusing on semantic information" (l. 313).
2. The paper narrative is very focused on hallucinations and factuality, but there is not too much analysis that shows whether the gains are indeed related to hallucinations, or that the attention interventions relate specifically to model hallucination behaviors, as opposed to a general degradation/improvement in generation quality. One option in my view is to change the framing of the paper to be a bit more generic, that would also work better with parts like §5.6 that talk about generation quality and not hallucination. An alternative approach would be to keep the current narrative but add analyses and examples that show some specific connection to model hallucinations. I thought the analysis in §5.3 is good in this respect, ideally I would have liked to see more like that throughout the paper (e.g., analyses across datasets, examples of base model hallucinations and how they are resolved with the method etc.).
3. Regarding the scaling factor $\alpha$ (Eq. 2 / Eq. 5) I think something does not add up here with the formulation. In the original contrastive decoding formulation, we have $log(p_{orig}) - log(p_{mod})$, which is equivalent to $log(p_{orig}/p_{mod})$ - thus the score of each token reflects the _ratio_ between the probabilities $p_{orig}$ and $p_{mod}$ for this token. In my opinion multiplying just one side of the subtraction by a factor is not a straightforward way to scale the ratio/contrast, and moreover achieves the opposite effect from the one described in the text: $(1+\alpha)*log(p_{orig})$ is equivalent to $log(p_{orig}^{1+\alpha})$, and since $p<1$, an $\alpha$ larger than one actually gives *less* weight to $p_{orig}$ relative to standard CD, not more.

**Questions:**

Questions:
1. How were the hyperparameters $th_a$ and $th_b$ chosen? Was there a development/test split? Were the same parameter values used for the different benchmark datasets? Also re the layer interval choice - the values are specified in the paper but I am not sure how they were selected.
2. Do you have an explanation for the poor performance of the other approaches on Factor? The numbers here for DoLA and SLED here are very different from those reported in the SLED paper for the same models.

Additional comments/suggestions:
* Another relevant work re the attention on different tokens is Yu et al. 2024, "Unveiling and Harnessing Hidden Attention Sinks". Also some classic interpretability works deal with attention on punctuation, parts of speech, etc., e.g., Vig & Belinkov 2019. "Analyzing the Structure of Attention in a Transformer Language Model".
* Most of the citations are not in the right format (latex \cite{} should be switched to \citep{})
* Ideally colors should be consistent between Fig. 3a and 3b
* There is not enough spacing below the caption of Figure 3
* §5.6: it can be more clearly phrased that the GPT-5 nano results are only in the appendix

Typos:
- l. 150 over-allocating focus -> over-allocate focus
- l. 207 observe several trends and patterns emerge -> observe several trends and patterns that emerge
- Figure 3b Vallina -> Vanilla

---

> ### Author Response · Authors · 2025-11-22
> **Response to Reviewer cria (1/2)**
>
> Dear Reviewer cria,
>
> Thank you for your valuable suggestions and questions. Here is our response:
>
> **Q1:Choice of Hyperparameters**
>
> Our work involves several hyperparameters, including $α$, $th_a$, $th_b$, $β$, and the layer ranges. These parameters fall into two categories. Among them, $th_a$, $th_b$ and the layer ranges are closely related to the intrinsic characteristics of the model. In our experiments, they are determined based on the attention distribution patterns of the model. The remaining parameters, $α$ and $β$, are mainly associated with contrastive decoding.
>
> We analyzed attention weights of different models on tasks with varying input lengths. The table below reports the top-3 attention values of P-type tokens on the 2nd and 10th layers—representative mid-layers of the first and second stages—as well as the summed C-type attention for LLaMA2-7B, LLaMA3-8B, and Mistral-7B on representative cases from StrategyQA, NQ, and HaluEval-Sum. Across inputs of 64 (NQ), 438 (StrategyQA), and 1672 (HaluEval-Sum) tokens, P-type tokens consistently show a sharp drop between the high-weight tokens and the rest. Based on this, we set $th_a$=0.1 for computational convenience and keep it fixed in all experiments.
>
> **Layer 2**
>
> | Model   | Type    | StrategyQA               | HaluEval-Sum            | NQ                     |
> |---------|---------|---------------------------|--------------------------|-------------------------|
> | llama2  | P top3  | 0.462 / 0.350 / 0.019     | 0.534 / 0.315 / 0.012    | 0.473 / 0.407 / 0.020   |
> |         | C sum   | 0.037                     | 0.044                    | 0.028                   |
> | llama3  | P top3  | 0.817 / 0.022 / 0.016     | 0.804 / 0.005 / 0.001    | 0.897 / 0.020 / 0.004   |
> |         | C sum   | 0.043                     | 0.135                    | 0.045                   |
> | mistral | P top3  | 0.580 / 0.220 / 0.009     | 0.552 / 0.242 / 0.009    | 0.650 / 0.235 / 0.010   |
> |         | C sum   | 0.069                     | 0.072                    | 0.044                   |
>
> **Layer 10**
> | Model   | Type    | StrategyQA               | HaluEval-Sum            | NQ                     |
> |---------|---------|---------------------------|--------------------------|-------------------------|
> | llama2  | P top3  | 0.262 / 0.220 / 0.053     | 0.205 / 0.197 / 0.053    | 0.390 / 0.292 / 0.038       |
> |         | C sum   | 0.092                     | 0.155                    | 0.175                   |
> | llama3  | P top3  | 0.358 / 0.090 / 0.068     | 0.238 / 0.045 / 0.037    | 0.670 / 0.012 / 0.005      |
> |         | C sum   | 0.133                     | 0.505                    | 0.270                   |
> | mistral | P top3  | 0.324 / 0.103 / 0.066     | 0.325 / 0.092 / 0.063    | 0.390 / 0.111 / 0.063       |
> |         | C sum   | 0.093                     | 0.171                    | 0.279                   |
>
> For $th_b$, we set it to 0.05 in Table 1, while in the subsequent experiments it is set to 0. In fact, we found that this parameter has little practical impact. We will remove it in the revised version and update the corresponding results in Table 1 using $th_b$ = 0. Table 1 will be updated as:
>
> | Model       | Method | TQA MC1 | TQA MC2 | TQA MC3 | TQA-v2 MC1 | TQA-v2 MC2 | TQA-v2 MC3 | StrategyQA| HellaSwag| Factor |
> |-------------|--------|---------|---------|---------|-------------|-------------|-------------|-------|-------|--------|
> | LLaMA2-7B   | Greedy | 34.18   | 60.44   | 32.62   | 26.71       | 41.88       | 19.15       | 60.96 | 75.68 | 63.56  |
> |             | DoLa   | 33.42   | 64.22   | 31.30   | 27.22       | 42.54       | 19.28       | 60.61 | 74.55 | 47.03  |
> |             | SLED   | 35.06   | 63.87   | 32.65   | 23.67       | 47.24       | 22.78       | 61.31 | 67.83 | 52.54  |
> |             | Ours   | 37.72   | 66.74   | 38.14   | 28.73       | 45.85       | 21.13       | 62.66 | 79.09 | 67.80  |
> |  LLaMA2-13B  | Greedy | 34.68   | 64.01   | 32.59   | 27.72       | 43.14       | 19.64       | 66.07 | 79.12 | 67.80  |
> |             | DoLa   | 29.87   | 63.25   | 30.96   | 29.87       | 48.85       | 24.35       | 65.55 | 49.21 | 44.92  |
> |             | SLED   | 35.19   | 65.02   | 32.68   | 24.18       | 46.98       | 23.35       | 66.81 | 71.68 | 64.83  |
> |             | Ours   | 39.62   | 65.98   | 38.73   | 30.89       | 46.70       | 22.06       | 68.38 | 81.86 | 70.34  |
> | LLaMA3-8B   | Greedy | 34.68   | 64.06   | 33.27   | 29.75       | 48.51       | 22.92       | 67.88 | 79.69 | 66.95  |
> |             | DoLa   | 34.18   | 64.58   | 32.90   | 29.75       | 48.45       | 22.97       | 67.42 | 79.69 | 55.51  |
> |             | SLED   | 35.95   | 65.35   | 33.23   | 26.20       | 50.09       | 24.72       | 67.60 | 73.14 | 63.98  |
> |             | Ours   | 37.97   | 67.30   | 39.99   | 33.67       | 52.40       | 25.24       | 68.25 | 82.97 | 73.73  |

---

> ### Author Response · Authors · 2025-11-22
> **Response to Reviewer cria (2/2)**
>
> The choice of layer ranges can be referenced from Figure 3(a) in the paper. We use the growth trend of the summed C - type attention as the key factor in determining this range. These parameters remain consistent across models and different tasks.
>
> For $β$, we follow standard practices in contrastive decoding and fix it to 0.1. The remaining parameter $α$ is adjustable across different tasks. In all experiments conducted after updating Table 1, we set α = 1 by default for all tasks except Factor and TruthfulQA. For TruthfulQA, since our implementation follows the publicly released code of DoLa and SLED, we set $α$ = 0 to remain consistent with their setup. In addition, we conducted supplementary experiments following the updated version of DoLa, where $α$ is set to 1. The resulting performance is shown in updated Table 1, marked as TruthfulQA-v2.
>
> These findings indicate that $α$ = 1 is generally suitable for most tasks, with the exception of Factor, where we set $α$ = 0.5.
> We also conducted an ablation study on $α$, and the results are presented below:
>
> | Model     | α   | TQA-v2 MC1 | TQA-v2 MC2 | TQA-v2 MC3 | StrategyQA| NQ EM | NQ F1 | HotpotQA EM | HotpotQA F1 | Factor |
> |-----------|-----|-------------|-------------|-------------|--------|-------|-------|--------------|-------------|--------|
> | llama2-7b | 0.5 | 30.76       | 51.86       | 25.70       | 63.93 | 23.75 | 18.75 | 19.30        | 17.34       | 67.80  |
> |           | 1   | 28.73       | 45.85       | 21.13       | 62.66 | 21.58 | 17.55 | 18.93        | 17.02       | 62.71  |
> |           | 1.5 | 27.22       | 42.66       | 18.91       | 62.66 | 20.05 | 16.37 | 18.35        | 17.47       | 60.59  |
>
> We observe that $α$ = 0.5 and $α$ = 1 work well for most tasks. Although our method involves several hyperparameters, in practice they do not require careful per - task tuning, as the stable performance across these $α$ values demonstrates.
>
> **Q2:Performance on the FACTOR dataset**
>
> We rely on the publicly available SLED implementation, but we found that its performance on the EXPERT-FACTOR is suboptimal. We attempted to tune several parameters, however, the performance could not be improved to a satisfactory level.
>
> **Q3:Explanation of Model Hallucination**
>
> In the case study presented in Section 3.1, after blocking the model’s C-type tokens, the output shifts from the correct answer “baseball” to the incorrect answer “sumo wrestling.” We interpret this behavior as a form of induced hallucinations, where intervening on the model’s internal attention causes it to generate an incorrect response, and further suppress potential hallucinations through contrastive decoding methods. This is also why our experiments include several datasets grounded in factual and knowledge-intensive content, as such datasets are well suited to probing the model’s understanding of knowledge-rich information, thereby enabling a rigorous evaluation of our hallucination-mitigation approach.
>
> **Q4:Clarification on the Role of α in the Formulation**
> Thank you for pointing out this issue. The explanation of the role of α in Eq. 2 / Eq. 5 was problematic and may have led to misunderstanding. In our formulation, α is merely a heuristic weight that adjusts the relative contribution of the original model’s score. We will revise the paper to clarify this point and eliminate the misleading interpretation in the current version.
>
> Thank you for pointing out all the detailed issues in our paper. We will address and correct them one by one in the subsequent revision. We also appreciate the related papers you mentioned, which are highly relevant to our work.

---

> ### Comment · Reviewer_cria · 2025-11-25
>
> Thank you for your responses to my comments and questions.
> In terms of hyperparameters there is an improvement here, removing $th_b$ is definitely a step in the right direction. I admit I do not quite understand how the analysis you posted here leads to a choice of $th_a=0.1$, but in any case keeping it fixed for all experiments is also a good thing. So on the plus side it seems there are fewer parameters to tune and they are relatively stable (but on the minus side the method for choosing the parameters is currently still not stated clearly in the paper text, which is very important).
>
> Unfortunately, I do not feel my other concerns have been seriously addressed. Specifically:
> * The various vague motivating statements that are not backed up by concrete experiments/citations.
> * The connection between the method and the theme of hallucinations remains tenuous, beyond the choice of datasets this connection is assumed throughout the paper text but in my mind is never convincingly demonstrated.
> * Regarding the way $\alpha$ is used: I do not find it particularly productive to respond that it is "merely a heuristic weight" - that is clear to me, but also heuristic weights should preferably have some logic behind them. As I mentioned, the current formulation of $log(p_{orig}^{1+\alpha})-log(p_{sub})$ is not a very standard or intuitive way to control the relative contribution between two log-probability values.

---

> > ### Author Response · Authors · 2025-12-03
> > **Response to Reviewer cria**
> >
> > Dear Reviewer cria,
> >
> > Thank you for your response. Here is our response:
> >
> > **Q1:Choice of hyperparameters**
> >
> > The reason for setting $th_a=0.1$ is that we observed the distribution of P-type tokens to be relatively uneven, and using 0.1 as the threshold helps reduce the number of intervened tokens.
> >
> > **Q2:Explanation of hallucinations**
> >
> > Regarding model hallucination, our focus aligns with the definition of factuality hallucination discussed in [1]. Factuality hallucination refers to the divergence between the generated content and established real-world facts, typically manifesting as factual contradictions or fabrications. It may include both intrinsic and extrinsic hallucinations. Such errors arise mainly from the probabilistic nature of language models, which tend to prioritize coherence and fluency over factual correctness.
> >
> > In our study, most of the selected datasets provide well-defined ground-truth answers, enabling us to reliably determine whether a model’s output is factually accurate. This allows us to evaluate how effectively a model mitigates hallucination through empirical results.
> >
> > **Q3:Explanation of the motivating statements**
> >
> > l. 278
> >
> > From the results shown in Figure 3(b), we observe that the model’s performance drops significantly under certain attention intervention strategies. Combined with our earlier explanation of hallucinations, we interpret this degradation as a form of hallucination induction, where the intervention leads the model to produce poorer and less reliable outputs.
> >
> > l. 300
> >
> > We argue that excessive attention to certain semantically rich tokens is more likely to induce hallucinations. In [2], the authors observe that when multiple conditions are present, LLMs tend to prioritize the condition with higher semantic salience. Semantically informative tokens naturally carry stronger associative patterns and therefore more easily overshadow other constraints, increasing the likelihood of generating content that is contextually plausible but incorrect.
> >
> > l. 313
> >
> > We argue that the attention distribution itself can serve as a form of model explanation. This perspective is supported by a growing body of attention-based interpretability research, such as [3] [4] [5], which demonstrate that attention patterns provide meaningful insights into how Transformer models process and prioritize information. When the model gradually assigns more attention to C-type tokens, this shift can be interpreted as the model beginning to focus on semantic information.
> >
> > **Reference**
> >
> > [1] A survey on hallucination in large language models: Principles, taxonomy, challenges, and open questions
> >
> > [2] Knowledge Overshadowing Causes Amalgamated Hallucination in Large Language Models
> >
> > [3] Attention is Not Not Explanation
> >
> > [4] Quantifying Attention Flow in Transformers
> >
> > [5] The Role of Attention Mechanisms in Enhancing Transparency and Interpretability of Neural Network Models in Explainable AI

---

### Official Review · Reviewer_8mPs · 2025-10-28

**Soundness:** 3
**Presentation:** 3
**Contribution:** 3
**Rating:** 6
**Confidence:** 4

**Summary:**

The paper proposes LayerCake, a training-free decoding method that builds a contrastive next-token distribution by targeted attention interventions at specific layers and on specific token types. Empirically, the authors observe that punctuation-like “structural” tokens dominate early layers, while “conceptual” tokens carry semantics in mid layers. Experiments on LLaMA-2/3, Mistral, and Qwen across several QA benchmarks show consistent improvements prior work on contrastive-decoding like DoLa.

**Strengths:**

Constructing a contrastive signal by purposefully inducing erroneous predictions through targeted interventions is an elegant idea. The intervention design, which selectively suppresses attention to specific token types at their most influential layers, creates a meaningful contrastive distribution that exposes how factual reasoning emerges within the model. The link between token category (e.g. structural vs. conceptual) and layer range (early vs. mid vs. late) seems well-motivated, and the resulting perturbed distributions seem to provide a good way to reweight base logits toward more truthful outputs.

The method is seems simple to deploy. It requires no finetuning or architectural modifications, operates entirely at decode time, and generalizes across different model families. This practicality adds to its contribution. It can be adopted immediately with existing checkpoints and standard inference stacks.

Empirically, the work is supported by a broad evaluation over diverse QA-style benchmarks and several model scales. The improvements are consistent and are complemented by a series of interpretability analyses that link changes in attention allocation to model behavior. The “structural → semantic grounding → consolidation → final alignment” pattern offers an interpretable mental model for where factual reasoning resides within the network. This staged progression provides a solid rationale for why the specific intervention sites are effective.

**Weaknesses:**

Most evaluations focus on short-form, question-answering style tasks, which raises questions about the generality of the approach. It remains unclear whether the approach would perform equally well for more open-ended forms of text generation such as long-form summarization, creative writing, or code synthesis. These tasks involve richer discourse structures, longer contexts, and a more complex interplay of coherence and factuality than typical QA settings. The specific intervention strategy: emphasizing punctuation in early layers and conceptual tokens in mid layers; may implicitly rely on the structure of QA prompts. Extending the method to settings with different input-output formats or reasoning demands would significantly strengthen the generality claim.

The reliance on heuristic token classification and manually chosen thresholds introduces another limitation. Decisions such as which tokens to suppress and where to intervene across depth are defined through fixed rules rather than being learned or inferred from the model’s own behavior. This could limit robustness and portability across architectures or tokenizers. A more systematic sensitivity analysis, or ideally an adaptive mechanism that infers token-layer associations directly from attention statistics or model internals could be an interesting extension of the work.

From an engineering perspective, the approach introduces nontrivial inference overhead. The contrastive decoding step requires two forward passes per token, effectively doubling compute time. Practical deployment would probably require techniques like batched contrastive evaluation, layer caching, or partial reuse of intermediate activations. A more detailed discussion of such optimizations would help establish the approach’s feasibility for real-world use.

Finally, it would be valuable to disentangle the contribution of the interpretability-guided design from the general contrastive decoding framework. Comparing the proposed strategy against variants that apply random or uninformed interventions, e.g., such as suppressing attention at arbitrary layers or token subsets, would reveal whether the improvement truly arises from interpretability insights or merely from the presence of a contrastive perturbation.

**Questions:**

How well does the approach generalize beyond QA to settings like summarization, long-form reasoning, or code generation?

Could the token categories or layer boundaries be discovered adaptively, for example from attention statistics or intermediate states rather than fixed heuristics?

How sensitive are results to prompt style or language? Would the method still work if structural cues like punctuation were removed or heavily altered?

The authors can improve my opinion about the work by convincingly addressing my questions and weaknesses raised.

---

> ### Author Response · Authors · 2025-12-03
> **Response to Reviewer 8mPs**
>
> Dear Reviewer cria,
>
> Thank you for your valuable suggestions and questions. Here is our response:
>
> **Q1:Explanation of the prompt structure**
>
> Although we evaluate our method on a wide range of QA-related datasets, these datasets vary substantially in input length—from fewer than one hundred tokens (e.g., NQ), to several hundred tokens (e.g., StrategyQA), and even over one thousand tokens (e.g., HaluEval-Sum).
>
> Moreover, the prompt structures are not limited to a single QA format. For example, GSM8K uses a typical question–answer template such as:
>
> "Q: If there are 3 cars in the parking lot and 2 more arrive, how many cars are there now?
> A: …"
>
> In contrast, HellaSwag adopts a continuation-style setup, where the model simply completes a sentence without an explicit QA prompt, such as:
>
> "Roof shingle removal: A man is sitting on a roof …"
>
> HaluEval-Sum uses yet another format involving document-summary-judgement triplets, such as:
>
> "Document: …
> Summary: …
> Your judgement: …"
>
> These examples illustrate that our evaluation already spans diverse prompt structures and a wide range of input lengths. In future work, we plan to test our method on an even broader set of datasets.
>
> **Q2:Adaptive Mechanism**
>
> Thank you for the suggestion. An adaptive mechanism is indeed an important direction for our future work. Based on the current results, the patterns we inferred from the model and the corresponding rule-based interventions already generalize well across different model architectures and tokenizers. In future work, we plan to further explore adaptive intervention strategies to reduce the number of hyperparameters and extend our approach to a broader range of models.
>
> **Q3:Inference Overhead**
>
> In Section 5.5 of the paper, we discuss the latency overhead. By checking in advance whether the candidate set contains sufficient alternative tokens, we can decide whether contrastive decoding is necessary. This allows us to avoid the additional cost of contrastive decoding in steps where the model’s output is already highly certain. In future work, we will explore additional strategies to further improve efficiency and minimize the extra computational overhead.
>
> **Q4:Random Intervention Comparison**
>
> Based on Llama2-7B, we randomly intervened on 10%, 20%, and 50% of the tokens in a set of randomly selected layers—specifically layers 4, 7, 10, 14, 19, 23, and 28. The results are shown in the table:
>
> | Intervention Ratio | NQ (EM / F1)  | HellaSwag | StrategyQA |
> |--------------------|---------------|-----------|------------|
> | 0.1                | 17.98 / 15.49 | 75.61     | 61.44      |
> | 0.2                | 18.78 / 16.09 | 74.89     | 60.79      |
> | 0.5                | 19.86 / 17.18 | 68.49     | 61.35      |
> | Ours               | 21.58 / 17.55 | 79.09     | 62.66      |
>
> As can be seen, the effect of random interventions is limited, which further demonstrates the effectiveness of our strategy.

---

### Official Review · Reviewer_A2x8 · 2025-10-31

**Soundness:** 2
**Presentation:** 2
**Contribution:** 1
**Rating:** 2
**Confidence:** 4

**Summary:**

This paper is another contrastive decoding paper that resembles the idea of DoLa to improve factuality. They didn't contrast between layers, instead, they weaken attention to punctuation tokens in early layers and to conceptual tokens in middle layers to make a messed-up output distribution. And the do contrastive decoding with the original final layer output distribution and the messed-up output distribution. It seems works on models like LLaMA, Mistral, and Qwen, improving factual accuracy by around 3-6%.

**Strengths:**

- Experiment results on models like LLaMA, Mistral, and Qwen seems to improve factual accuracy on several benchmarks.

**Weaknesses:**

- The paper introduced so many hyperparams: α、tₕₐ、tₕ_b、β、layer ranges in the method, however, it's unclear how the authors find the hyperparams. The only clear thing is:

> We set β = 0.1 throughout the paper.

Otherwise, the authors mentioned that

> thresholds tₕₐ, tₕ_b, and α are determined empirically

 but there is no explanation or details for it. What do you mean by "determined empirically"? It's possible that the author is adjusting the hyperparams based on each individual test set performance, which will be not allowed. If it's not the case, please provide the way that you select the hyperparams, for example, list the dev set performance and prove that you're choosing the best hyperparams on dev set and it transfers well to the test test. I suspect that there could be sensitivity in hyperparams between different tasks as the method is generally complex than the previous decoding method, especially when there are so many parameters to tune, so it may not be generalizing well.
- The method requires access to the attention weights and modifying the attention weights, which is not feasible for optimized attention layer like FlashAttention2, which is commonly used in inference engines. In practice it maybe not be very useful in real production. In contrast, methods like DoLa doesn't need access or modifying attention weights.
- The author should replace all \citet{} into \citep{} when citing refereneces, it's not standard to use \citet everywhere and it affects the reading experiences.

**Questions:**

Can you analytically show how you determine the thresholds tₕₐ, tₕ_b, and α? It's not valid to say vague things like "determined empirically" in a research paper. If it was tuned on the test set, it's not allowed. If it's tuned on the dev set, show me the numbers on the dev set and the details as a proof.

---

> ### Author Response · Authors · 2025-11-22
> **Response to Reviewer A2x8 (1/2)**
>
> Dear Reviewer A2x8,
>
> Thank you for your valuable suggestions and questions. Here is our response:
>
> **Q1:Choice of Hyperparameters**
>
> Our work involves several hyperparameters, including $α$, $th_a$, $th_b$, $β$, and the layer ranges. These parameters fall into two categories. Among them, $th_a$, $th_b$ and the layer ranges are closely related to the intrinsic characteristics of the model. In our experiments, they are determined based on the attention distribution patterns of the model. The remaining parameters, $α$ and $β$, are mainly associated with contrastive decoding.
>
> We analyzed attention weights of different models on tasks with varying input lengths. The table below reports the top-3 attention values of P-type tokens on the 2nd and 10th layers—representative mid-layers of the first and second stages—as well as the summed C-type attention for LLaMA2-7B, LLaMA3-8B, and Mistral-7B on representative cases from StrategyQA, NQ, and HaluEval-Sum. Across inputs of 64 (NQ), 438 (StrategyQA), and 1672 (HaluEval-Sum) tokens, P-type tokens consistently show a sharp drop between the high-weight tokens and the rest. Based on this, we set $th_a$=0.1 for computational convenience and keep it fixed in all experiments.
>
> **Layer 2**
>
> | Model   | Type    | StrategyQA               | HaluEval-Sum            | NQ                     |
> |---------|---------|---------------------------|--------------------------|-------------------------|
> | llama2  | P top3  | 0.462 / 0.350 / 0.019     | 0.534 / 0.315 / 0.012    | 0.473 / 0.407 / 0.020   |
> |         | C sum   | 0.037                     | 0.044                    | 0.028                   |
> | llama3  | P top3  | 0.817 / 0.022 / 0.016     | 0.804 / 0.005 / 0.001    | 0.897 / 0.020 / 0.004   |
> |         | C sum   | 0.043                     | 0.135                    | 0.045                   |
> | mistral | P top3  | 0.580 / 0.220 / 0.009     | 0.552 / 0.242 / 0.009    | 0.650 / 0.235 / 0.010   |
> |         | C sum   | 0.069                     | 0.072                    | 0.044                   |
>
> **Layer 10**
> | Model   | Type    | StrategyQA               | HaluEval-Sum            | NQ                     |
> |---------|---------|---------------------------|--------------------------|-------------------------|
> | llama2  | P top3  | 0.262 / 0.220 / 0.053     | 0.205 / 0.197 / 0.053    | 0.390 / 0.292 / 0.038       |
> |         | C sum   | 0.092                     | 0.155                    | 0.175                   |
> | llama3  | P top3  | 0.358 / 0.090 / 0.068     | 0.238 / 0.045 / 0.037    | 0.670 / 0.012 / 0.005      |
> |         | C sum   | 0.133                     | 0.505                    | 0.270                   |
> | mistral | P top3  | 0.324 / 0.103 / 0.066     | 0.325 / 0.092 / 0.063    | 0.390 / 0.111 / 0.063       |
> |         | C sum   | 0.093                     | 0.171                    | 0.279                   |
>
> For $th_b$, we set it to 0.05 in Table 1, while in the subsequent experiments it is set to 0. In fact, we found that this parameter has little practical impact. We will remove it in the revised version and update the corresponding results in Table 1 using $th_b$ = 0. Table 1 will be updated as:
>
> | Model       | Method | TQA MC1 | TQA MC2 | TQA MC3 | TQA-v2 MC1 | TQA-v2 MC2 | TQA-v2 MC3 | StrategyQA| HellaSwag| Factor |
> |-------------|--------|---------|---------|---------|-------------|-------------|-------------|-------|-------|--------|
> | LLaMA2-7B   | Greedy | 34.18   | 60.44   | 32.62   | 26.71       | 41.88       | 19.15       | 60.96 | 75.68 | 63.56  |
> |             | DoLa   | 33.42   | 64.22   | 31.30   | 27.22       | 42.54       | 19.28       | 60.61 | 74.55 | 47.03  |
> |             | SLED   | 35.06   | 63.87   | 32.65   | 23.67       | 47.24       | 22.78       | 61.31 | 67.83 | 52.54  |
> |             | Ours   | 37.72   | 66.74   | 38.14   | 28.73       | 45.85       | 21.13       | 62.66 | 79.09 | 67.80  |
> |  LLaMA2-13B  | Greedy | 34.68   | 64.01   | 32.59   | 27.72       | 43.14       | 19.64       | 66.07 | 79.12 | 67.80  |
> |             | DoLa   | 29.87   | 63.25   | 30.96   | 29.87       | 48.85       | 24.35       | 65.55 | 49.21 | 44.92  |
> |             | SLED   | 35.19   | 65.02   | 32.68   | 24.18       | 46.98       | 23.35       | 66.81 | 71.68 | 64.83  |
> |             | Ours   | 39.62   | 65.98   | 38.73   | 30.89       | 46.70       | 22.06       | 68.38 | 81.86 | 70.34  |
> | LLaMA3-8B   | Greedy | 34.68   | 64.06   | 33.27   | 29.75       | 48.51       | 22.92       | 67.88 | 79.69 | 66.95  |
> |             | DoLa   | 34.18   | 64.58   | 32.90   | 29.75       | 48.45       | 22.97       | 67.42 | 79.69 | 55.51  |
> |             | SLED   | 35.95   | 65.35   | 33.23   | 26.20       | 50.09       | 24.72       | 67.60 | 73.14 | 63.98  |
> |             | Ours   | 37.97   | 67.30   | 39.99   | 33.67       | 52.40       | 25.24       | 68.25 | 82.97 | 73.73  |

---

> ### Author Response · Authors · 2025-11-22
> **Response to Reviewer A2x8 (2/2)**
>
> The choice of layer ranges can be referenced from Figure 3(a) in the paper. We use the growth trend of the summed C - type attention as the key factor in determining this range. These parameters remain consistent across models and different tasks.
>
> For $β$, we follow standard practices in contrastive decoding and fix it to 0.1. The remaining parameter $α$ is adjustable across different tasks. In all experiments conducted after updating Table 1, we set α = 1 by default for all tasks except Factor and TruthfulQA. For TruthfulQA, since our implementation follows the publicly released code of DoLa and SLED, we set $α$ = 0 to remain consistent with their setup. In addition, we conducted supplementary experiments following the updated version of DoLa, where $α$ is set to 1. The resulting performance is shown in updated Table 1, marked as TruthfulQA-v2.
>
> These findings indicate that $α$ = 1 is generally suitable for most tasks, with the exception of Factor, where we set $α$ = 0.5.
> We also conducted an ablation study on $α$, and the results are presented below:
>
> | Model     | α   | TQA-v2 MC1 | TQA-v2 MC2 | TQA-v2 MC3 | StrategyQA| NQ EM | NQ F1 | HotpotQA EM | HotpotQA F1 | Factor |
> |-----------|-----|-------------|-------------|-------------|--------|-------|-------|--------------|-------------|--------|
> | llama2-7b | 0.5 | 30.76       | 51.86       | 25.70       | 63.93 | 23.75 | 18.75 | 19.30        | 17.34       | 67.80  |
> |           | 1   | 28.73       | 45.85       | 21.13       | 62.66 | 21.58 | 17.55 | 18.93        | 17.02       | 62.71  |
> |           | 1.5 | 27.22       | 42.66       | 18.91       | 62.66 | 20.05 | 16.37 | 18.35        | 17.47       | 60.59  |
>
> We observe that $α$ = 0.5 and $α$ = 1 work well for most tasks. Although our method involves several hyperparameters, in practice they do not require careful per - task tuning, as the stable performance across these $α$ values demonstrates.
>
> **Q2:Use of FlashAttention2**
>
> We conducted preliminary tests on datasets such as StrategyQA and NQ, and observed that when the input length is only a few hundred tokens, the inference speed shows no significant difference between using and not using FlashAttention2. Since we have removed the parameter $th_b$, the original processing can now be implemented in FlashAttention2 by introducing an attention mask, achieving the same effect without obtaining the complete attention weights in the second stage.
>
> We will include additional comparisons in the revised version to show the performance difference depending on whether FlashAttention2 is used, if these comparisons can help address your concerns.
>
> Thank you again for your valuable suggestions and corrections. We will address these issues in our subsequent revisions.

---

### Author Response · Authors · 2025-12-03
**Summary of Discussion Points for the Area Chair**

Dear AC,

We sincerely appreciate your time and effort in handling our submission. We also thank all four reviewers for their thoughtful assessments and for recognizing the novelty, clarity and broad applicability of LayerCake. Here is the summary of discussion points during the rebuttal period:

- The reviewers’ concerns focused on whether the parameters introduced by our method require repeated task-specific tuning and how they should be selected (A2x8, cria, RPM2), whether the approach is constrained by fixed prompt formats (8mPs), the engineering feasibility of the intervention (A2x8, 8mPs), the clarity of the connection to hallucination and the underlying motivation (cria, RPM2), and the interpretation of the formulas used in the paper (cria).

- In our rebuttal, we made the following efforts to address these concerns: (1) We removed unnecessary hyperparameters and unified the remaining ones across all datasets except FACTOR—including the updated TruthfulQA evaluation—to demonstrate cross-task generality. We also provided empirical evidence motivating our hyperparameter choices. (2) We listed the input formats used in our experiments to highlight their diversity and reported the range of input lengths to show that our method is not limited to short-context inputs. (3) After eliminating extra hyperparameters, part of the intervention can be implemented through attention masking, making it compatible with FlashAttention2 and providing additional room for engineering-level speedups. (4) We added citations on hallucination to clarify the relationship between our method and hallucination mitigation and referenced relevant work to strengthen the motivation behind our design choices. (5) We corrected the phrasing that caused misunderstandings in the formula description and clarified that the hyperparameters are intended to control the contrast intensity.

We have incorporated all clarifications, analyses, and updated results into the revision. With these improvements, our work is now more robust in both motivation and empirical support.

Thank you again for taking the time to read our summary.

Best regards,

Authors of Submission 13304

---

### Meta-Review · Area_Chair_4php · 2026-01-06

**Summary:**

This paper proposes a training-free contrastive decoding method that selectively suppresses attention to punctuation and conceptual tokens at specific layers to induce a degraded distribution for contrast. There are still some major concerns about hyperparameter selection, generalization, and method clarification.

For the benefit of this paper, we regretfully recommend rejection. Note that this is not a discouragement. The authors are encouraged to address these concerns, and we believe the paper has the potential to become a strong future submission.

**Reviewer Concerns:**

While the authors addressed concerns about the removal of the specific $\gamma$ hyperparameter and provided clarifications on FlashAttention compatibility, some major concerns still remain. Specifically:

- Hyperparameter Sensitivity and Heuristics (Reviewers A2x8, Reviewers cria, Reviewers RPM2): The most critical outstanding issue is the reliance on multiple hyperparameters ($\alpha, \beta$, layer ranges).
  - Reviewer A2x8 noted that determining thresholds "empirically" is vague and potentially invalid if tuned on test sets. This has not been explicitly answered in the rebuttal.
  - Reviewer cria remained unconvinced by the rebuttal, noting that the method is complex and lacks an overarching logic for parameter selection.
  - Reviewer RPM2 pointed out inconsistencies in the updated tables regarding $\alpha$, which makes the experimental results feel less confident.

- Method clarification (Reviewers cria, Reviewers RPM2):
  - Reviewer cria explicitly stated that the connection between the proposed attention interventions and the mitigation of hallucinations remains tenuous and was not convincing in the rebuttal.
  - Reviewer RPM2 questioned the necessity of token-level classification over simple attention-value suppression, suggesting the complexity may not be justified.

- Generalization (Reviewer 8mPs): While Reviewer 8mPs was more positive, they noted that the reliance on fixed rules rather than adaptive mechanisms limits robustness. The concern regarding the method's applicability beyond short-form QA (e.g., to long-context or code generation) remains a validity constraint.

**Reviewer Scores:**

All the reviewers are likely to maintain current score.

---

### Decision · Program_Chairs · 2026-01-26

Reject